# The Role of Diet Quality in Mediating the Association between Ultra-Processed Food Intake, Obesity and Health-Related Outcomes: A Review of Prospective Cohort Studies

**DOI:** 10.3390/nu14010023

**Published:** 2021-12-22

**Authors:** Samuel J. Dicken, Rachel L. Batterham

**Affiliations:** 1Centre for Obesity Research, Department of Medicine, University College London (UCL), London WC1E 6JF, UK; samuel.dicken.20@ucl.ac.uk; 2Bariatric Centre for Weight Management and Metabolic Surgery, University College London Hospital (UCLH), London NW1 2BU, UK; 3National Institute for Health Research, Biomedical Research Centre, University College London Hospital (UCLH), London W1T 7DN, UK

**Keywords:** obesity, diet, ultra-processed food, NOVA classification, diet quality, dietary pattern, non-communicable disease

## Abstract

Prospective cohort studies show that higher intakes of ultra-processed food (UPF) increase the risk of obesity and obesity-related outcomes, including cardiovascular disease, cancer and type 2 diabetes. Whether ultra-processing itself is detrimental, or whether UPFs just have a lower nutritional quality, is debated. Higher UPF intakes are inversely associated with fruit, vegetables, legumes and seafood consumption. Therefore, the association between UPFs and poor health could simply be from excess nutrient intake or from a less healthful dietary pattern. If so, adjustment for dietary quality or pattern should explain or greatly reduce the size of the significant associations between UPFs and health-related outcomes. Here, we provide an overview of the literature and by using a novel approach, review the relative impact of adjusting for diet quality/patterns on the reported associations between UPF intake and health-related outcomes in prospective cohort studies. We find that the majority of the associations between UPFs, obesity and health-related outcomes remain significant and unchanged in magnitude after adjustment for diet quality or pattern. Our findings suggest that the adverse consequences of UPFs are independent of dietary quality or pattern, questioning the utility of reformulation to mitigate against the obesity pandemic and wider negative health outcomes of UPFs.

## 1. Introduction

Obesity (defined as an excess accumulation of fat that may result in adverse health [1]) is a leading cause of poor health, increasing the risk of non-communicable disease (NCD), all-cause mortality and negatively impacting on quality of life [2,3,4]. Management strategies for obesity prevention and treatment are therefore important.

Diet has long been a cornerstone of weight management, with dietary policies being a core feature of government and health organisation strategies to reduce obesity worldwide. Indeed, poor diets are a leading cause of preventable obesity-related death and NCD, including cancer, cardiovascular disease (CVD) and type 2 diabetes (T2DM), accounting for 11 million deaths annually [5,6]. As such, dietary improvements could prevent one in every five deaths [5]. There is converging evidence that a healthy diet consists predominantly of whole, plant-based foods, including fruit, vegetables, pulses, nuts, whole grains and oily fish [7,8,9,10,11,12,13,14]. Such diets, as exemplified by the Mediterranean diet and Dietary Approaches to Stop Hypertension (DASH), are high in fibre and limit saturated fat, sodium and added sugar intake. In contrast, Western diets high in refined grains, red and processed meat, sweets and sugar-sweetened beverages are rich in saturated fat, sodium and added sugar and associated with an increased risk of disease [13,14,15,16,17].

Despite the importance of specific nutrients and food groups within overall dietary patterns for health, it is becoming increasingly clear that other dimensions of diets are important [18]. In recent decades, a nutrition transition has resulted in a global shift away from consuming minimally processed foods, and towards ultra-processed alternatives [19,20], away from home-prepared dishes, and towards ready-to-eat meals and snacks [21]. This same period has seen a rapid rise in the global prevalence of obesity in children and adults [22]. Besides their nutrient content, healthy dietary patterns such as the Mediterranean diet tend to be minimally processed [15], and unhealthy dietary patterns such as the Western diet tend to be ultra-processed [11,16,23].

Whether ultra-processed diets are detrimental to health simply because they are of a poor nutritional quality, or whether the nature and extent of processing itself has health consequences is an ongoing debate [24]. Several recent systematic reviews, meta-analyses and reviews have discussed the prevalence of UPF consumption and its impact on health-related outcomes. However, no reviews to date have considered how dietary adjustment in prospective cohort studies may alter the significance and magnitude of effect estimates. This review provides a brief overview of the current state of the literature as well as the current key discussion points regarding mechanisms of action, before reviewing in detail the prospective analyses adjusting for dietary quality, which provides important insights into the relative role of nutrient content compared with ultra-processing on obesity risk and adverse health-related outcomes.

## 2. NOVA Classification

Several classification systems have been developed to categorise food and drink based on levels of processing, including the International Food Information Council, International Agency for Research on Cancer and NOVA classifications [25]. The most commonly used is the NOVA classification, which considers the nature, extent and purpose of processing, not the act of processing itself, to be important [26]. The NOVA food classification consists of four groups: minimally processed foods (MPF), processed culinary ingredients (PCI), processed foods (PF) and ultra-processed foods (UPF) (Table 1) [27]. UPFs are industrial formulations, typically with five or more ingredients including additives, flavourings and colours that no longer resemble their original constituent ingredients [28]. Nutritional quality, such as nutrients to limit content, is not a core aspect of the NOVA classification.

In recent decades, the contribution of UPFs to diets worldwide has been increasing year on year [29]. In the US and UK, over 55% of the average daily energy intake now comes from UPFs, and those in the highest quintiles consume over 75% of their daily energy intake from UPFs [30]. Additionally, UPFs are becoming increasingly more prevalent in the diets of infants, children and adolescents [31,32].

## 3. UPFs, Obesity Risk and Health-Related Outcomes

Systematic reviews and meta-analyses of prospective cohort studies and cross-sectional studies show that UPF consumption is associated with an increased risk of weight gain, overweight and obesity [33,34,35,36,37,38], as well as other obesity-related health outcomes [33,34], including hypertension, type 2 diabetes (T2DM) [38,39], cancer [33], cardiovascular disease (CVD) [33,34], depression and all-cause mortality [33,35,36,37,40]. In Europe, a 1% increase in the national household availability of UPFs is associated with a 0.25% increase in the national prevalence of obesity, after adjusting for income, physical inactivity and smoking [41]. Additionally, increases in ultra-processed food and drink volume sales per capita are associated with population-level BMI trajectories [42]. The rising contribution of UPFs to diets worldwide poses a significant threat to addressing the obesity epidemic.

## 4. Mechanisms and Current Debates around Ultra-Processing: Correlation or Causation?

There is increasing evidence showing that UPFs are linked with obesity and other adverse health-related outcomes. However, the potential mechanisms that lead to these adverse health outcomes are diverse, and still largely debated (readers are directed to other comprehensive overviews for further reading on potential mechanisms [43,44,45]). These mechanisms can be broadly considered as being as a result of nutrient content, or as a result of ultra-processing [43].

From a nutrient perspective, UPFs have on average a higher energy density (2.3 vs. 1.1 kcal/g) and lower nutrient density than minimally processed foods [44,46]. UPFs tend to be high in saturated fat, added sugar and sodium [47], with meta-analyses demonstrating that diets higher in UPFs tend to contain greater intakes of total energy, free sugars and total and saturated fat, and lower intakes of fibre, protein and some micronutrients [30,48]. The high palatability of UPFs has the potential to promote a faster eating rate and energy overconsumption [44], with daily energy intake increasing as the proportion of daily energy intake from UPFs increases (3.47 kcal increase per 1% increase in daily energy intake from UPFs) [30].

However, aspects of ultra-processing may also increase the risk of obesity and other adverse health-related outcomes. Textural and structural changes to the food matrix as a result of ultra-processing can also allow for UPFs to be consumed more quickly [49,50,51]. Reducing the oro-sensory exposure (OSE) time of a food can delay the onset of satiation [52], and UPFs have been shown to be less satiating than minimally processed foods [53,54]. The delayed satiation from faster eating rates can promote increased energy intake [55]. Food matrix changes can also alter nutrient bioavailability, and the harm from UPFs may come from the fact that they tend to be more hyperglycaemic than MPFs [53,54]. Besides the nutritional quality of UPFs and degradation of the food matrix, the additive content and excessive heat treatment of UPFs have also been proposed to lead to changes in gut microbiota and promote inflammation [56,57].

Beyond nutrients and ultra-processing, behavioural aspects of UPFs and the local, environmental and systemic drivers influencing food choice are also important [58]. The heavy marketing [59,60], low cost [46], high availability [61] and large portion sizes of UPFs [43,62] can make them preferable choices over minimally processed options.

## 5. UPF Removal or UPF Reformulation: The Case for ‘Healthy’ UPFs?

Actions to reduce the risks associated with UPF intake have largely been either reformulation to improve the nutrient profile of UPFs, or avoidance of UPFs altogether. Whether experts support UPF reformulation or UPF avoidance is dependent on the views regarding which mechanisms link UPFs with poor health.

Both those in favour of limiting UPFs [63], and those against the NOVA classification [64,65,66], acknowledge that the nutritional quality of UPFs is an important factor. Even some proponents of NOVA and reducing UPF intake have suggested that the saturated fat, added sugar and sodium content of UPFs is important, despite this not being a core aspect of the UPF definition [28]. For example, authors have focussed on the impact of reducing UPF intake on changes in saturated fat, added sugar and sodium intake and dietary quality, and the subsequent benefit of these changes on disease risk [67,68,69,70]. Critics of NOVA/supporters of reformulation argue that any link between UPFs and adverse health is solely due to their nutrient content; that some UPFs are just high in saturated fat, added sugar and sodium and that some UPFs are not nutritionally inferior, with some studies showing no difference in saturated fat, added sugar and sodium intakes across extremes of UPF intake [64,71,72].

Indeed, many UPFs are nutritionally poor and energy dense, but not all are. Studies demonstrate that UPFs tend to relate with existing nutrient profile indices, based on saturated fat, added sugar and sodium content [73]. In comparison with the Nutri-Score (ranking foods from class A to E, where A is high and E is low nutritional quality) used across several European countries, the majority of UPFs are class C, D or E, whereas the majority of unprocessed or minimally processed foods are class A or B [74]. However, 26% of class A foods are UPFs, largely being UPF ready meals or dairy products. Studies comparing UPFs with other nutrient indices (such as the Nutrient Rich Foods index, based on the protein, fibre, vitamins, minerals, saturated fat, added sugar and sodium content of food) show similar findings; most UPFs are low in nutritional quality, but some are high, and most MPFs are high in nutritional quality, but some are still low in quality [46]. Indeed, a range of UPFs have been identified as being ‘healthy’, based on nutrient profiling [65]. ‘Healthy’ UPFs are often reformulations and plant-based alternatives [65,74], which carry nutritional claims such as ‘fat free’, ‘reduced salt’, ‘low sugars or ‘added fibre’ according to European Food Safety Authority guidelines [75]. Other ‘healthy’ UPFs such as fortified bread have been suggested to be important sources of vitamins and minerals [64,65,76], and avoidance of such UPFs may result in micronutrient deficiencies [77]. Therefore, two foods can be defined as having a high level of nutritional quality, but with very different levels of processing [78].

Given that particular UPFs, such as reformulations, can be considered to be of a similar or greater nutritional quality than some MPFs, it has been suggested that these UPFs are therefore healthy and nutritious [65,76]. Experts proposing that reformulations are sufficient to address all issues surrounding UPFs are making the assumption that the association between UPF intake and adverse health is mediated solely by their content of specific nutrients [71]. Experts proposing avoidance of all UPFs and arguing that reformulations are insufficient to significantly improve health are making the assumption that no UPFs can be considered to be healthy [79]. Such experts argue that reformulation does not address aspects of ultra-processing [80,81,82]; reformulated UPFs still have a degraded food matrix [83,84], and components of the raw constituent foods are still lost [85].

In summary, there is agreement that energy dense foods high in saturated fat, sodium and added sugar are harmful to health and should be limited. Such foods also tend to be ultra-processed, but not all are [73,74]. Despite the mounting evidence showing the adverse impacts of UPFs, the argument between nutrients and ultra-processing, and therefore between reformulation or avoidance of UPFs, is ongoing [24,81,82]. Further research understanding the relative impact of nutrients vs. ultra-processing is therefore warranted. However, largely overlooked to date, is the fact that many published prospective cohort studies have already considered the overlap between nutrition and ultra-processing, performing dietary adjustments of models to delineate the association between UPF intake, obesity and adverse health-related outcomes.

## 6. Review of Prospective Studies Adjusting for Dietary Quality

One of the main criticisms against the NOVA classification is that UPFs simply capture nutrient poor foods high in saturated fat, sodium and added sugar [71,73]. Furthermore, it is well established that the overall dietary pattern is important for health [10]. Higher UPF intakes are inversely associated with MPF intake, including fruit, vegetables, cereals, beans, legumes and seafood intake [30]. Therefore, the association between high UPF intake and poor health could simply be from excess nutrient intake, or from a less healthful dietary pattern. If this were the case, adjustment for participants’ dietary saturated fat, sugar and sodium intake, or adjustment for their overall dietary pattern should explain the significant associations found between higher intakes of UPF and adverse health-related outcomes in prospective cohort studies, either rendering the association to be non-significant, or greatly reducing the size of the association.

Many prospective studies in adults have performed dietary adjustments, with only a small proportion not adjusting for aspects of dietary quality [86,87,88,89,90,91]. A greater proportion of prospective studies during gestation [92,93], or in children [94,95,96,97,98,99,100,101], have not performed dietary adjustments. These dietary adjustments can be broadly classed as adjustment for fat (typically saturated fat), carbohydrate (typically sugar) and sodium, adjustment for dietary patterns (including Mediterranean diet, Healthy Eating Index (HEI), Alternate Healthy Eating Index (AHEI), Dietary Guidelines for Americans Adherence Index (DGAI), healthy and Western dietary patterns and Food Standards Agency Nutrient Profiling System Dietary Index (FSA-NPS-DI)), or other dietary adjustments (typically for specific food groups such as fruit and vegetables).

Table 2 presents the 37 longitudinal, observational studies that report some form of adjustment for diet quality/pattern in their analyses investigating the association between UPF intake as defined by NOVA, and health-related outcomes (the search process and criteria for the review is detailed in the Appendix A). Table 2 also presents the association between UPF intake and health-related outcomes from adjusted models preceding the dietary adjustment, or where not reported, the adjusted model including the dietary adjustment.

Across 37 studies, 87 health-related outcomes were assessed using 91 models. Of the 66 models that demonstrate a significant association between UPF and a health-related outcome, 64 remained significant following adjustment for diet quality or diet pattern. In total, 136/142 dietary adjustments did not explain the significance of the association between UPF intake and a health-related outcome. Across four studies, all four models demonstrated higher UPF intakes were significantly associated with all-cause mortality [102,103,104,105]. No dietary adjustments (15/15) altered the significance of UPF intake with all-cause mortality. Across 13 models within five studies, 11 were significantly associated with a CVD outcome [104,105,117,118,119]. 29/31 dietary adjustments did not alter the significance of UPF intake with CVD outcomes. Across two studies, UPF intake was significantly associated with cancer outcomes in 2/5 models [105,126]. 8/8 dietary adjustments did not alter the significance of the association between UPF intake and the two cancer outcomes. In two models significantly associated with T2DM, 7/7 dietary adjustments did not alter the significance [123,124]. Across nine studies, 23/26 models demonstrated a significant association between UPF intake and adult and child anthropometrics (weight/body mass index (BMI)/fat mass index (FMI) gain, other measures of adiposity and risk of overweight/obesity) [106,107,108,109,110,111,112,113,115]. 40/43 dietary adjustments did not alter the significance of these associations.

### 6.1. Adjustment for Saturated Fat, Sugar and Sodium, and for Dietary Pattern

Table 3 presents the adjustments for saturated fat, sodium and added sugar. Table 4 presents the adjustments for dietary pattern. All but one study retained the significant association between UPF and the health-related outcome after adjustments for saturated fat, sodium and added sugar intake. All but two studies retained the significant association between UPF and the health-related outcome after adjustment for dietary pattern.

Within the NutriNet-Santé cohort, several studies have performed dietary adjustments for the associations between UPF intake and all-cause mortality, CVD, overweight/obesity incidence, T2DM, cancer and functional gastrointestinal disorders [102,106,117,124,126,127,129]. Schnabel et al. found a 15% (95% confidence interval: 1.04, 1.27) increased risk of all-cause mortality per 10% increase in UPF intake in the diet [102]. Adjusting for French dietary guideline adherence or for both French dietary guideline adherence and for Western dietary pattern still resulted in each 10% increment in UPF intake being associated with a 14% (1.04, 1.27) or 19% (1.05, 1.35) increased risk, respectively, of all-cause mortality [102].

Srour et al. reported a 12% (1.05, 1.20), 13% (1.02, 1.24) and 11% (1.01, 1.21) increased risk of CVD, coronary heart disease (CHD) and cerebrovascular disease, respectively, per 10% increase in UPF in the diet [117]. Multiple dietary adjustments did not alter these risk estimates. First, adjusting for saturated fat, sodium and added sugar intake resulted in a 13% (1.05, 1.20), 14% (1.03, 1.26) and 12% (1.02, 1.22) increased risk of CVD, CHD and cerebrovascular disease, respectively [117]. Second, adjusting instead for a healthy dietary pattern still resulted in an 11% (1.03, 1.19), 11% (1.00, 1.23, *p* = 0.04) and 10% (1.00, 1.20, *p* = 0.04) increased risk of CVD, CHD and cerebrovascular disease, respectively [117]. Third, adjusting for intakes of sugary products, red and processed meat, salty snacks, beverages, fats and sauces also still resulted in a 12% (1.04, 1.20), 12% (1.01, 1.24) and 11% (1.01, 1.22) increased risk of CVD, CHD and cerebrovascular disease, respectively, per 10% increase in UPF in the diet [117].

In a separate study, Srour et al. reported a 15% (1.06, 1.25) increased risk of T2DM with each 10% increase in UPF in the diet, which included adjustment for dietary quality using the FSA-NPS-DI [124]. Again, subsequent dietary adjustments did not alter the increased risk of T2DM. A 10% increase in UPF in the diet was still associated with a 19% (1.09, 1.30) increased risk when further adjusting for saturated fat, sodium, sugar and dietary fibre intake, a 13% (1.04, 1.24) increased risk after adjusting for healthy and Western dietary patterns, and a 14% (1.04, 1.25) increased risk after adjusting for intakes of red and processed meat, sugary drinks, fruits and vegetables, whole grains, nuts, and yogurt in place of the FSA-NPS-DI adjustment [124]. Srour et al. also adjusted for absolute amounts of unprocessed or minimally processed food intake, which few studies have performed to date. This adjustment also did not alter the increased risk of T2DM (hazard ratio (HR) per 100g/day increase in UPF intake: 1.05 (1.02, 1.08) [124].

Fiolet et al. reported a 12% (1.06, 1.18) and 11% (1.02, 1.22) increased risk of all cancer and breast cancer, respectively, per 10% increase in UPF in the diet [126]. Adjustment for lipids (including fat), sodium and carbohydrate intake had no impact on the risk of all cancer (HR: 1.12 (1.07, 1.18)) or breast cancer (HR: 1.11 (1.01, 1.21)) per 10% increase in UPF in the diet, respectively [126]. Adjustment instead for Western dietary pattern also did not change the 12% (1.06, 1.18) and 11% (1.02, 1.22) increased risks [126].

Beslay et al. reported a greater BMI gain (β: 0.02 kg/m^2^ (0.01, 0.02)) and increased risk of overweight (HR: 1.11 (1.08, 1.14)) or obesity (HR: 1.09 (1.05, 1.13)), per 10% increase in UPF in the diet [106]. Adjusting for healthy and Western dietary patterns did not alter the greater BMI gain (β: 0.02 kg/m^2^ (0.01, 0.02)), or increased risk of overweight (HR: 1.10 (1.07, 1.13)) or obesity (HR: 1.11 (1.07, 1.15)), and neither did adjustment for saturated fat, sugar, sodium and dietary fibre intake, which also resulted in a greater BMI gain (β: 0.02 kg/m^2^ (0.01, 0.02)), and increased risk of overweight (HR: 1.10 (1.08, 1.13)) or obesity (HR: 1.10 (1.06, 1.14), per 10% increase in UPF intake [106].

Schnabel et al. identified an increased risk of irritable bowel syndrome (IBS) (odds ratio (OR): 1.24 (1.12, 1.38)) and functional dyspepsia (OR: 1.26 (1.07, 1.48)) when comparing the highest vs. lowest quartiles of UPF intake [129]. Adjustment for adherence to French dietary guidelines did not alter the increased risk of IBS (OR: 1.25 (1.12, 1.39)) or functional dyspepsia (OR: 1.25 (1.05, 1.47)) across extreme quartiles of UPF intake [129].

Four studies within the Seguimiento Universidad de Navarra (SUN) cohort have adjusted for fat, added sugar and sodium intake, or for dietary pattern. Rico-Campa et al. demonstrated that the highest vs. lowest quartile of UPF intake was associated with a 62% (1.13, 2.33) increased risk of all-cause mortality [103]. Adjustment for saturated and trans fats, added sugar and sodium intake still resulted in a 69% (1.12, 2.56) increased risk of all-cause mortality. A 58% (1.10, 2.28) increased risk still remained after adjusting for Mediterranean diet pattern adherence instead [103].

Llavero-Valero et al. reported that the highest vs. lowest tertile of UPF intake was associated with a 53% (1.06, 2.22) increased risk of T2DM, which was unaltered (HR: 1.50 (1.02, 2.21)) after adjusting for Mediterranean diet pattern adherence [123].

Gómez-Donoso et al. found a 41% (1.15, 1.73) increased risk of incident depression in the highest vs. lowest quartile of UPF intake, which was still associated with a 33% (1.07, 1.64) higher risk of incident depression after further adjustment for other covariates, including Mediterranean diet pattern adherence [132].

Leone et al. identified an increased risk of gestational diabetes in females aged 30 and over (OR 1st vs. 3rd tertile: 2.05 (1.03, 4.07)), which was unaltered after adjustment for Mediterranean diet pattern adherence (OR 1st vs. 3rd tertile: 2.06 (1.05, 4.06)) [114].

In the US Third National Health and Nutrition Examination Survey (NHANES III) cohort, there was a 31% (1.09, 1.58) increased risk of all-cause mortality in the highest vs. lowest quartile of UPF intake, which remained significant after further adjustment for dietary quality score using the HEI-2000 (*p*-trend = 0.001) [104]. However, diet-adjusted risk estimates were not provided.

In the Italian Moli-sani cohort, the highest vs. lowest quartile of UPF intake was associated with a 32% (1.15, 1.53) higher risk of all-cause mortality, 65% (1.29, 2.11) higher risk of CVD mortality, and a 63% (1.19, 2.25) higher risk of ischemic heart disease (IHD)/cerebrovascular mortality [105]. Adjusting for saturated fat, sugar, sodium and dietary cholesterol intake resulted in a 28% (1.09, 1.49), 56% (1.19, 2.03) and 33% (0.94, 1.90) increased risk of all-cause, CVD and IHD/cerebrovascular mortality, respectively, in the highest vs. lowest quartile of UPF intake. Bonaccio et al. also individually adjusted for saturated fat, sugar, sodium and dietary cholesterol in turn, with UPF intake remaining significantly associated with all-cause, CVD and IHD/cerebrovascular mortality in all adjustments, except for sugar intake and IHD/cerebrovascular mortality (HR: 1.37 (0.98, 1.90)). Adjusting instead for Mediterranean diet pattern adherence resulted in a 26% (1.09, 1.46), 58% (1.23, 2.03) and 52% (1.10, 2.09) increased risk of all-cause, CVD and IHD/cerebrovascular mortality [105].

In the Framingham Offspring cohort, each additional serving of UPF per day was associated with a 5% (1.02, 1.08), 9% (1.02, 1.16), 7% (1.03, 1.12) and 9% (1.04, 1.15) increased risk of overall CVD, CVD mortality, hard CVD and hard coronary heart disease, respectively [118]. Further adjustment for diet quality using the DGAI-2010 still resulted in a 4% (1.01, 1.07), 9% (1.02, 1.16), 6% (1.02, 1.11) and 9% (1.03, 1.15) increased risk of overall CVD, CVD mortality, hard CVD and hard coronary heart disease [118].

In the US Prostate, Lung, Colorectal, and Ovarian Cancer Screening Trial cohort, the highest vs. lowest quintile of UPF intake was associated with a 50% (1.36, 1.64) increased risk of CVD mortality, and a 68% (1.50, 1.87) increased risk of heart disease mortality [119]. Multiple dietary adjustments did not alter this risk; adjustment for saturated fat, added sugar and sodium resulted in a 48% (1.34, 1.63) and 65% (1.47, 1.85) increased risk of CVD mortality and heart disease mortality, adjustment for diet quality using HEI-2005 resulted in a 48% (1.35, 1.63) and 67% (1.49, 1.86) increased risk of CVD mortality and heart disease mortality, and adjustment instead for red meat, processed meat, whole grains, fruit, vegetables, fibre and dairy intake also still resulted in a 49% (1.35, 1.64) and 66% (1.48, 1.86) increased risk of CVD mortality and heart disease mortality, respectively [119].

In the European Prospective Investigation into Cancer and Nutrition (EPIC) cohort, each additional standard deviation (SD) increment in UPF intake per day was associated with a 0.12 kg (0.09, 0.15) greater increase in weight over 5 years of follow-up, which was unaltered after further adjusting for Mediterranean diet score (β: 0.12 kg/5 years (0.09, 0.15)) [111]. In sensitivity analyses of fully adjusted models including Mediterranean diet adherence, UPF intake was associated with a higher risk of overweight/obesity (relative risk (RR): 1.05 (1.04, 1.06)) and obesity (RR: 1.05 (1.03, 1.07)) per 1SD increase in UPF per day. This corresponded to a 15% (1.11, 1.19) higher risk of overweight/obesity in participants with normal weight and a 16% (1.09, 1.23) higher risk of obesity in participants with overweight at baseline, when comparing the highest vs. lowest quintiles of UPF intake [111].

In the China Nutrition and Health Survey (CNHS), consuming ≥50 g of UPF per day was associated with an increased risk of overweight/obesity (OR: 1.85 (1.58, 2.17)) and central obesity (OR: 2.04 (1.79, 2.33)), when compared to no UPF intake. Adjustment for traditional and modern dietary patterns did not alter the increased risks (overweight/obesity, OR: 1.45 (1.21, 1.74), central obesity, OR: 1.50 (1.29, 1.74)) [108].

In the Seniors Study on Nutrition and Cardiovascular Risk in Spain (Seniors-ENRICA-1), Sandoval-Insausti et al. found an increased risk of abdominal obesity (OR: 1.62 (1.04, 2.54) in the highest vs. lowest tertile of UPF intake, which was unaltered after adjustment for Mediterranean diet adherence and fibre and omega-3 fatty acid intake (OR: 1.61 (1.01, 2.56)) [110].

Donat-Vargas et al. identified an increased risk of hypertriglyceridaemia (OR: 2.00 (1.04, 3.85)) and low-HDL cholesterol (OR: 2.04 (1.22, 3.41)), as well as a significant increase in blood triglycerides (β: 6.11 mg/dL (1.30, 10.91)) when comparing the highest vs. lowest tertile of UPF intake [136]. Adjustment for unprocessed or minimally processed food intake did not alter the increased risk of hypertriglyceridaemia (OR: 2.66 (1.20, 5.90)), low-HDL cholesterol (OR: 2.23 (1.22, 4.05)) or change in blood triglycerides (β: 6.87 mg/dL (1.48, 12.27)) [136].

In the Pelotas-Brazil 2004 Birth Cohort, Costa et al. found a 0.09 kg/m^2^ (0.07, 0.10) greater gain in FMI from ages 6 to 11, per 100 g daily increase in UPF intake [116]. Adjustment for other NOVA food groups (minimally processed and processed food, and processed culinary ingredients intake) significantly increased the associated FMI gain to 0.14 kg/m^2^ (0.13, 0.15) from age 6 to 11, per 100 g daily increase in UPF intake [116].

In the Avon Longitudinal Study of Parents and Children (ALSPAC) cohort, the highest vs. lowest quintile of UPF intake was associated with a 0.06 kg/m^2^ (0.04, 0.08) and 0.03 kg/m^2^ (0.01, 0.05) greater yearly increase in BMI and FMI, respectively, from the age of 7 to 24 [115]. Adjustment for saturated fat, sugar, sodium and fibre intake did not alter the association between UPF intake and increases in BMI (β: 0.07 kg/m^2^/year (0.04, 0.08)) or FMI (β: 0.03 kg/m^2^/year (0.01, 0.05)) [115].

Koniecnzna et al. conducted a prospective analysis of the PREDIMED-Plus trial over the course of 12 months. Each 10% increment in UPF intake was associated with increases in total (β: 0.09 (0.06, 0.13)) and visceral (β: 0.09 (0.05, 0.13)) fat mass z-scores. Adjusting for overall repeated measures of saturated and trans fat, sodium, glycaemic index, alcohol and fibre intake across the 12 month study did not alter the significant association between UPF intake and increases in total (β: 0.06 (0.03, 0.09) and visceral (β: 0.06 (0.01, 0.10)) fat mass z-scores per 10% increase in daily UPF intake [109]. Adjusting instead for overall repeated measures of Mediterranean diet pattern adherence across the 12 month study also did not alter the association between each 10% increment in UPF intake and increases in total (β: 0.06 (0.02, 0.09)) and visceral (β: 0.06 (0.01, 0.10)) fat mass z-scores [109].

In the Tianjin Chronic Low-grade Systemic Inflammation and Health (TCLSIH) Cohort Study, the highest vs. lowest quartile of UPF intake was associated with a 17% (1.07, 1.29) higher risk of non-alcoholic fatty liver disease (NAFLD) in the age, sex and BMI adjusted model. After adjustment for other confounders including for a healthy diet score based on fruit, vegetable, red meat and fish intake, the increased risk associated with the highest vs. lowest quintile of UPF intake was 19% (1.08, 1.31) [125].

Zhang et al. found a 21% (1.10, 1.33) increased risk of hyperuricaemia in the highest vs. lowest quartile of UPF intake, which was still associated with a 17% (1.06, 1.30) increased risk of hyperuricaemia after adjustment for dietary pattern [134].

In a separate study, Zhang et al. reported that each 10% increment in UPF in the diet was associated with a −0.30 kg (−0.50, −0.09) and −0.0043 kg/kg weight (−0.0073, −0.0014) yearly reduction in absolute and weight-adjusted grip strength, respectively [138]. Adjustment for further covariates including a healthy diet score (based on fruit, vegetable, unprocessed red meat and fish intake), dietary supplement use and protein and milk intake did not alter the association, with each 10% increment in UPF intake still associated with −0.32 kg (−0.53, −0.11) and −0.0046 kg/kg weight (−0.0076, −0.0016) yearly reductions in absolute and weight-adjusted grip strength, respectively [138].

In a combined analysis of the Nurses’ Health Study, the Nurses’ Health Study II and the Health Professionals Follow-up Study, Lo et al. found a 75% (1.29, 2.35) increased risk of Crohn’s disease in the highest vs. lowest quartile of UPF intake after adjusting for age, cohort and calendar year. The increased risk was unchanged after further covariate adjustments, including for diet quality defined by the AHEI-2010 (HR: 1.70 (1.23, 2.35)) [130].

In the Prospective Urban Rural Epidemiology (PURE) cohort, Narula et al. identified an 82% (1.22, 2.72) increased risk of inflammatory bowel disease (IBD) and a 450% (1.67, 12.13) increased risk of Crohn’s disease in those consuming five or more UPF servings per day, compared with those consuming less than one serving per day. Adjustment for AHEI-2010 still resulted in a 92% (1.28, 2.90) increased risk of IBD and a 490% (1.78, 13.45) increased risk of Crohn’s disease [128].

In the Norwegian Mother, Father and Child Cohort Study, Borge et al. reported that each 1 SD increase in maternal UPF intake was associated with an increase in absolute (0.38 (0.27, 0.49)) and relative (4.5% (3.3, 4.9)) measures of child attention deficit hyperactivity disorder (ADHD) symptoms at age 8, using the Parent Rating Scale for Disruptive Behaviour Disorders [137]. Adjustment for child Diet Quality Index (based on diet diversity, diet quality and diet equilibrium [139]) did not alter the associated increase in absolute (0.25 (0.13, 0.38)) or relative (3.0% (1.5, 4.5)) ADHD symptoms [137].

Three studies have considered the impact of diet quality and dietary pattern using alternative methods. In the ATTICA cohort, each additional weekly serving of UPF was associated with a 10% (1.02, 1.21) increased risk of CVD. Kouvari et al. then performed sub-group analysis based on Mediterranean diet pattern adherence. Participants with moderate to high adherence to the Mediterranean diet had an attenuated (8% (0.98, 1.19)) risk of CVD per weekly serving of UPF, whereas participants with low adherence to the Mediterranean diet had an even greater risk of 19% (1.12, 1.25), per weekly serving of UPF [140].

Bonaccio et al. identified that diet quality (defined by the FSA-NPS-DI) was only significantly associated with all-cause mortality in high UPF consumers (HR per 1 SD increase in FSA-NPS-DI: 1.14 (1.05, 1.25), but not in low UPF consumers (HR: 1.00 (0.93, 1.07) (*p* for interaction = 0.034) in the Moli-sani cohort [141]. The interaction between diet quality and UPF intake was not significant for CVD mortality.

In the ENRICA study, the highest vs. lowest quartile of UPF intake had a 44% (1.01, 2.07) increased risk of all-cause mortality [91]. Instead of dietary adjustment, Blanco-Rojo et al. compared the highest vs. lowest intakes of nutrients from UPF intake, including total, saturated and trans fat, carbohydrates, sugar, sodium and fibre [91]. The nutrient content of UPFs was not associated with an increased mortality risk, except for trans fat (HR highest vs. lowest quartile: 1.39 (1.00, 1.92), *p* = 0.047).

### 6.2. Adjustment for Fat, Sodium, Carbohydrate and Dietary Pattern

Two studies have simultaneously adjusted for fat, sodium and carbohydrate intake and for dietary pattern, which are reported in Appendix A. For cancer outcomes, Fiolet et al. adjusted for both intakes of lipids (including fat), sodium, and carbohydrates and Western dietary pattern, resulting in a 13% (1.07, 1.18) and 11% (1.01, 1.21) increased risk of all cancer and breast cancer per 10% increase in UPF in the diet [126].

Adjibade et al. identified a 21% (1.15, 1.27) higher risk of depressive symptoms per 10% increase in UPF in the diet in the NutriNet-Santé cohort [131]. After adjusting for intakes of lipids (including fat), sodium, and carbohydrates and for healthy and Western dietary patterns, the risk of depressive symptoms per 10% increase in UPF in the diet was still 22% (1.16, 1.29) [131].

### 6.3. Adjustment for Fat And/or Sugar and/or Sodium

Some studies have adjusted for one or two components of fat and/or sugar and/or sodium intake, rather than all three components. One study adjusted for carbohydrate intake, rather than sugar intake [114]. These adjustments are reported in Appendix A. The significant associations between UPF intake and all-cause mortality, overweight or obesity, central obesity, T2DM, hypertension, gestational weight gain, neonatal anthropometrics and blood lipid profiles were unchanged following these dietary adjustments [103,108,113,114,120,123,135,136].

### 6.4. Adjustment for Other Dietary Components

Other measures used for dietary adjustment are provided in Appendix A. Other dietary adjustments include for fried foods, fruit and vegetables, UPF soft drinks, multivitamin use and excluding bacon, sausage and processed meats from ultra-processed food intake. These adjustments had no impact on the association between higher intakes of UPF and risk of all-cause mortality, cancer, overweight/obesity, increased total and visceral fat mass, increased BMI and FMI, NAFLD, weight and waist circumference gain, adverse blood lipid profiles, grip strength decline, incident hypertension and renal function decline [103,104,106,107,109,111,112,115,121,122,125,126,133,136,138].

### 6.5. Dietary Adjustments That Explain the Association between UPF Intake and Health-Related Outcomes

To date, only two studies have performed dietary adjustments that explain the association between higher UPF intakes and adverse health-related outcomes. In the PREDIMED-Plus study, each 10% increase in UPF in the diet was associated with a 5% (0.00, 0.09, *p* = 0.031) increase in android:gynoid fat ratio z-score during 12 months of follow-up [109]. Adjusting for repeated measures of sodium, saturated and trans fat, alcohol, fibre and glycaemic index, or adjusting for repeated measures of Mediterranean Diet adherence during the 12-month follow-up period resulted in a non-significant association between UPF intake and android:gynoid fat ratio z-score [109].

In the Moli-sani cohort, the highest vs. lowest quartile of UPF intake had a 36% (1.01, 1.83) higher risk of other cause mortality (any mortality, excluding CVD and cancer). However, after adjusting for Mediterranean diet score, this became non-significant (1.26 (0.94, 1.69)) [105]. As noted in Section 6.1, the increased risk of IHD/cerebrovascular mortality also became non-significant after adjusting for saturated fat, sugar, sodium and dietary cholesterol.

### 6.6. Adjustment for Total Energy Intake

An ultra-processed diet has been shown to increase energy intake in comparison with a minimally processed diet [49]. Energy intake may be a mediator of both nutritional aspects (high energy density and palatability), and of some ultra-processing aspects (a degraded food matrix influencing oro-sensory exposure and satiety) of UPFs. Adjustment for total daily energy intake is not only useful to control for measurement error in epidemiological dietary assessment to improve risk estimation of other dietary measures [142,143], but it can also provide information on the associated risk between UPF intake and adverse health outcomes, independent of energy intake [144].

Adjustment for energy intake can be achieved using several methods [145,146]. However, it has typically been performed by energy-adjusting the UPF independent variable, either via the residual method (regressing UPF intake onto total energy intake to produce residuals) or via the nutrient density method (usually as ‘energy intake from UPFs/total energy intake’, though ‘total weight of UPFs/total food weight’ has also been used to capture the non-nutritive aspects of UPFs) [146]. Total energy intake is then included as a covariate in the model [146]. Some studies instead use absolute UPF intake as the independent variable, and then include total energy intake in the model.

Table 5 presents the prospective cohort studies performing adjustments for total energy intake. Forty-seven studies have performed some form of energy adjustment across 131 models. 80 models demonstrate a significant association between energy-adjusted UPF intake and a health-related outcome. 6/6 models were significantly associated with all-cause mortality, 12/15 models were significantly associated with any CVD outcome, 3/3 models were significantly associated with T2DM, 15/17 models were significantly associated with adult weight gain/overweight/obesity, and 15/25 models with gestational or child anthropometrics. Twenty-one non-significant models with energy adjustment were from multiple models for child appetitive traits (eight; Vedovato et al. [99]), childhood anthropometrics and glucose profiles (six; Costa et al. [98]), child asthma and wheezing (four; Machado Azeredo et al. [101]) and childhood lipid profiles (three; Rauber et al. [95]). Four studies provided insufficient detail on energy adjustments [87,96,140,147].

### 6.7. Prospective Studies Reporting Mediation Analyses

Besides being included as a covariate within models, formal mediation analysis can be used to determine whether dietary components mediate the association between UPF intake and adverse health-related outcomes [149,150]. Few studies to date have performed mediation analyses between UPF intake, dietary components and health-related outcomes.

Bonaccio et al. examined the mediating role of nutrients and energy content on all-cause mortality, CVD mortality and IHD/cerebrovascular mortality [105]. All dietary factors combined (sugar, saturated fat, dietary cholesterol, dietary sodium and energy content) significantly accounted for 41.3% ((11.9%, 78.5%), *p* < 0.001) of IHD/cerebrovascular mortality risk, but did not account for all-cause mortality (12.8% (1.6%, 56.5%), *p* = 0.14) or CVD mortality (11.5% (1.5%, 53.3%), *p* = 0.15) risk. Sugar content alone accounted for 23.2% ((9.7%, 45.9%), *p* < 0.001), 18.0% ((7.2%, 38.4%), *p* = 0.003) and 36.3% ((13.8%, 67.0%), *p* < 0.001)) of the associated risk between UPF intake and all-cause mortality, CVD mortality and IHD/cerebrovascular mortality, respectively. Saturated fat or sodium content did not account for any of the associated risks.

Fiolet et al. performed mediation analyses for sodium, total lipids, saturated fat, monounsaturated fat, polyunsaturated fat, carbohydrate and for Western dietary pattern, with all mediation effects for the association between UPF intake and overall cancer being less than 2% (all *p* > 0.05) [126].

Koniecnzna et al. found that repeated measures of saturated fat, trans fat and fibre explained 11–30% of the associations between UPF intake and increases in measures of central and overall adiposity over 12 months of follow-up [109]. Repeated measures of sodium, total energy intake and glycaemic index did not mediate any of the associations.

Costa et al. identified that 58.2% (0.07 kg/m^2^ (0.05, 0.10)) of the association between UPF intake and the increase in FMI from age 6 to 11 in children was mediated by energy content, with the remaining 41.8% being either a direct effect of ultra-processing, or as a result of unmeasured variables [116].

Vedovato et al. showed that energy intake was a mediator between UPF intake at 4 years of age and the appetite traits, ‘satiety responsiveness’ and ‘food fussiness’, but not with ‘food responsiveness’, at age 7 [99].

Gomes et al. showed that the percentage of total energy derived from UPFs in the third trimester was associated with total energy intake in the third trimester, which was also associated with gestational weight gain in the third trimester [93].

## 7. Discussion

This review provides novel insights into the relative impact of nutrient content and dietary patterns vs. ultra-processing on obesity and adverse health-related outcomes. The analyses reported here from prospective cohort studies have been largely unexplored to date. Consistent across many studies, adjustment for fat, sugar and sodium intake, or adjustment for adherence to a range of healthy or unhealthy dietary patterns has a minimal impact on the adverse associations between UPF intake and a diverse range of health-related outcomes. These findings strongly point towards aspects of ultra-processing as being important factors that impact health, and question the ability to conclude that the adverse outcomes from UPFs can be solely attributed to their nutritional quality.

A meta-analysis of nationally representative samples demonstrates that diets high in UPF tend to contain greater intakes of energy, free sugars, total and saturated fat, and lower intakes of fibre, protein and some micronutrients [30]. The NOVA classification therefore captures important aspects of nutrient quality, despite this not being a core aspect of the UPF definition [28]. It is unsurprising therefore, that the detrimental associations between UPF intake and obesity, CVD and all-cause mortality have been largely attributed to the poor nutritional quality of high UPF diets [71]. If this were the case, then adjustment for aspects of dietary quality should explain the associations between UPFs and poor health outcomes, or at least, explain a significant proportion of the association. However, the majority of the models from prospective studies retain a significantly increased risk of poor health from UPF intake, and are also largely unaltered in magnitude, following dietary adjustment. The findings from this review are in alignment with the results from a metabolic ward cross-over study, the only randomised controlled trial comparing diets of differing levels processing [49]. Participants consumed ad libitum, minimally processed or ultra-processed diets, matched for energy and nutrient content, for two weeks each. The ultra-processed diet resulted in greater energy intake (+508 ± 106 kcal/day), leading to weight gain (+0.9 ± 0.3 kg). In contrast, the minimally processed diet resulted in weight loss (−0.9 ± 0.3 kg), despite diets being matched for energy and nutrient content [49].

The Mediterranean diet, considered to be one of the healthiest dietary patterns for reducing CVD risk [151], consists predominantly of whole grains, fruits, vegetables, beans, pulses and legumes, of which, their consumption is inversely associated with UPF intake [30]. Therefore, the impact of UPFs on health could just be that they displace more healthful foods, or that they overlap with pre-established unhealthy dietary patterns. However, adjustment for Mediterranean diet adherence, for the Western dietary pattern or for other dietary pattern indices, did not alter the majority of the significant associations between UPF intake and health-related outcomes, including the increased risk of weight gain or obesity.

High UPF diets are also characterised by the displacement of minimally processed foods, as defined by NOVA [30]. Few studies have performed dietary adjustment for other NOVA food groups. However, in those that have, adjustment for other NOVA food groups not only did not explain, but in fact, increased the risk associated with UPF intake and FMI gain from age 6 to 11 [116], and adjustment for unprocessed or minimally processed food intake did not alter the increased risk of T2DM [124].

Although limited at this stage, these adjustments would suggest that UPF intake has a direct effect on health-related outcomes, rather than simply displacing healthy foods intake. This may indicate the importance of considering the nature and extent of food processing as an important dimension of dietary pattern analysis.

Discussions over the relative importance of nutrient content vs. ultra-processing continue [24,81]. However, recent reports have not taken into account the dietary adjustments from prospective studies reported in this review [24,81,152]. The aspects of ultra-processing that lead to adverse health outcomes are poorly understood, and the findings from this review highlight the need for research into mechanisms of ultra-processing as being a priority, in order to determine the long-term potential for UPF reformulation, or need for elimination to address the growing obesity pandemic. On a case-by-case basis, choosing UPF reformulations over high fat, salt or sugar alternatives can be beneficial to reduce intakes of nutrients to limit, which are known to be associated with poor health [153]. However, given the high prevalence of UPFs within diets [30], if ultra-processing itself directly results in poor health, then scaling up the case-by-case reformulation approach to the whole diet still leaves an ultra-processed dietary pattern that displaces minimally processed foods, and thus will not sufficiently address current health risks. The nutritional quality of food is important, but is not the sole determinant of the healthiness of a diet [10]. The importance of dietary patterns, food groups and foods as a whole, rather than specific individual nutrients, has previously been highlighted [8,154]. Indeed, the diversity of chemicals and nutrients consumed in human diets is vast, yet current nutrient profiling methods only consider a fraction of the 26,000 or so biochemicals in food [155].

Current dietary policies vary across nations and health organisations. The public health implications regarding whether UPFs should be reformulated based on nutrient content or removed from the diet are important. The UK government and Cancer Research UK currently adopt a reformulation approach to high fat, salt or sugar foods, and do not consider the nature and extent of processing in their dietary recommendations [156,157,158]. However, advice to avoid UPFs is becoming increasingly more prevalent. The American Heart Association now recommends limiting UPF intake [12], and the World Health Organisation and UNICEF recognise the importance of UPF consumption for ending childhood obesity [159,160]. UPFs are also recognised by the Pan American Health Organisation as important for reducing health risk, as part of their nutrient profiling model [161]. Some national dietary guidelines now encourage limiting UPF intake including Brazil [162], Uruguay [163] and Israel [164]. France is also planning to reduce UPF consumption by 20% from 2018 to 2021 [165].

This review discusses the results from over 1,000,000 participants across more than 20 different prospective cohorts, covering many countries, demographic profiles and age groups. The studies in this review utilise dietary assessment methods including 24-hour dietary records and food frequency questionnaires, that are not designed specifically for the application of NOVA classification. Similar foods can be classed as ultra-processed (e.g., pre-packaged bread) or processed (e.g., artisanal bread), which may result in the misclassification of foods with the dietary assessment methods used. Furthermore, other important dietary aspects may have not been captured or suitably adjusted for. Most studies have adjusted for fat, sugar and sodium intake or for overall dietary patterns, which are important dietary factors for health, and proposed to be explanatory factors for the associations between UPF intake and health outcomes.

## 8. Conclusions

Experts for and against the NOVA classification have often focussed on nutrient quality as an important explanatory link between UPFs, obesity and adverse health-related outcomes. However, many of the prospective studies published to date have performed analyses adjusting for nutrient content and overall dietary patterns. These adjustments do not explain the association between UPFs, obesity and adverse health-related outcomes, with estimates remaining significant. These findings raise important questions regarding current policy and future research needs, suggesting that the nature and extent of processing is an important dietary dimension, and whether UPF reformulations can sufficiently address the growing transition towards high UPF diets and the associated risk of obesity and poor health.

## Figures and Tables

**Table 1 nutrients-14-00023-t001:** Definition of NOVA classifications, from Monteiro et al., 2019 [28].

Group	Definition	Examples
1. Unprocessed and minimally processed foods	Unprocessed foods altered by processes such as the removal of inedible or unwanted parts, drying, crushing, grinding, fractioning, roasting, boiling, pasteurisation, refrigeration, freezing, placement in containers, vacuum packaging or non-alcoholic fermentation. Salt, sugar, oils or fats, or other food substances are not added. The primary aim is to extend the life of the food, enabling storage for longer use, and to make preparation easier or more diverse.	Fresh, squeezed, chilled, frozen, or dried fruit, leafy and root vegetables, brown rice, white rice, corn cob, beans, lentils, chickpeas, potatoes, sweet potatoes, mushrooms, meat, poultry, fish, seafood, meat cuts, eggs, fresh or pasteurised milk or plain yoghurt, fresh or pasteurised fruit or vegetable juices (with no added sugar, sweeteners or flavours), grits, flakes or flour made from corn, wheat, oats, or cassava, nuts and other oily seeds (with no added salt or sugar), herbs and spices used in culinary preparations, such as thyme, oregano and pepper, tea, coffee, and water.
2. Processed culinary ingredients	Substances derived from unprocessed and minimally processed foods, or from nature. They are created by industrial processes including pressing, centrifuging, refining, extracting or mining, and used in the preparation, seasoning and cooking of group 1 foods.	Oils and fats, sugar and salt.
3. Processed foods	Industrial products made by adding processed culinary ingredients found in group 2 to group 1 foods, using preservation methods such as canning and bottling. For breads and cheeses, non-alcoholic fermentation is used. Food processing in group 3 aims to increase the durability of group 1 foods and make them more enjoyable, by modifying or enhancing their sensory qualities.	Canned or bottled vegetables and legumes in brine, salted or sugared nuts and seeds, salted, dried, cured, or smoked meats and fish, canned fish (with or without added preservatives), fruits in syrup (with or without added antioxidants), freshly made unpackaged breads and cheeses.
4. Ultra-processed foods	Formulations of ingredients, mostly of exclusive industrial use, resulting from a series of industrial processes, many requiring sophisticated equipment and technology. Processes enabling the manufacture of ultra-processed foods include the fractioning of whole foods into substances, chemical modifications of these substances, assembly of unmodified and modified food substances using industrial techniques such as extrusion, moulding and pre-frying, frequent application of additives whose function is to make the final product palatable or hyper-palatable (‘cosmetic additives’), and sophisticated packaging, usually with synthetic materials.	Carbonated soft drinks, sweet or savoury packaged snacks, chocolate, confectionery, ice cream, mass-produced packaged breads and buns, margarines, biscuits, pastries, cakes, breakfast ‘cereals’, pre-prepared pies and pasta and pizza dishes, poultry or fish nuggets, sausages, burgers, hot dogs and other reconstituted meat products, powdered and packaged ‘instant’ soups, noodles and desserts.

**Table 2 nutrients-14-00023-t002:** Characteristics of prospective studies assessing the association between UPF intake and health-related outcomes whilst also reporting dietary adjustments.

Author, Year	Cohort	Sample	Country	Sample Size	Outcome	Method of Analysis	Effect Estimate (95%CI)
Schnabel 2019 [102]	Nutri-Net Santé	Adults ≥ 45	France	44,551	All-cause mortality	HR per 10% increase in UPF	1.15 (1.04, 1.27) ^1^
Rico-Campa 2019 [103]	SUN	University graduates	Spain	19,899	All-cause mortality	HR 1st vs. 4th quartile	1.62 (1.13, 2.33) ^2^
Kim 2019 [104]	NHANES III	Adults ≥ 20	US	11,898	All-cause mortality	HR 1st vs. 4th quartile	1.31 (1.09, 1.58) ^3^
				11,898	CVD mortality	HR 1st vs. 4th quartile	1.10 (0.74, 1.67) ^3^
Bonaccio 2021 [105]	Moli-sani	Adults	Italy	22,475	All-cause mortality	HR 1st vs. 4th quartile	1.32 (1.15, 1.53) ^4^
				22,475	Other cause mortality (exc. CVD and cancer)	HR 1st vs. 4th quartile	1.36 (1.01, 1.83) ^4^
				22,475	Cancer mortality	HR 1st vs. 4th quartile	1.00 (0.80, 1.26) ^4^
				22,475	CVD mortality	HR 1st vs. 4th quartile	1.65 (1.29, 2.11) ^4^
				22,475	IHD/cerebrovascular mortality	HR 1st vs. 4th quartile	1.63 (1.19, 2.25) ^4^
Beslay 2020 [106]	Nutri-Net Santé	Adults ≥ 18	France	110,260	BMI change (kg/m^2^)	Beta per 10% increase in UPF	0.02 (0.01, 0.02) ^5^
				55,307	Overweight	HR per 10% increase in UPF	1.11 (1.08, 1.14) ^5^
				71,871	Obesity	HR per 10% increase in UPF	1.09 (1.05, 1.13) ^5^
Mendonca 2016 [107]	SUN	Middle-aged University graduates	Spain	8451	Overweight/obesity	HR 1st vs. 4th quartile	1.26 (1.10, 1.35) ^6^
Li 2021 [108]	CHNS	Adults > 20	China	12,451	Overweight/obesity	OR none vs. ≥ 50g/day	1.85 (1.58, 2.17) ^7^
				12,451	Central obesity	OR none vs. ≥ 50g/day	2.04 (1.79, 2.33) ^7^
Koniecnzna 2021 [109]	PREDIMED-Plus	Adults aged 55–75 with overweight/obesity and metabolic syndrome	Spain	1485	Total fat mass (z-score)	Beta per 10% increase in UPF	0.09 (0.06, 0.13) ^8^
				1485	Visceral fat mass (z-score)	Beta per 10% increase in UPF	0.09 (0.05, 0.13) ^8^
				1485	Android:gynoid fat ratio (z-score)	Beta per 10% increase in UPF	0.05 (0.00, 0.09) (*p* = 0.031) ^8^
Sandoval-Insausti 2020 [110]	Seniors-ENRICA-1	Older adults	Spain	652	Abdominal obesity	OR 1st vs. 3rd tertile	1.62 (1.04, 2.54) ^9^
Cordova 2021 [111]	EPIC	Adults aged 25–70	Multi-national (nine countries)	348,748	Weight gain (kg)	Beta per 1SD increase in UPF/day	0.12 (0.09, 0.15) ^10^
				191,255	Overweight/obesity	RR per 1SD increase in UPF/day	1·05 (1·04, 1.06) ^10^
				103,259	Obesity	RR per 1SD increase in UPF/day	1·05 (1.03, 1.07) ^10^
Canhada 2020 [112]	ELSA-Brazil	Civil servants aged 35–74	Brazil	11,827	Large weight gain (≥90th percentile: ≥1.68 kg/year)	RR 1st vs. 4th quartile	1.27 (1.07, 1.50) ^11^
				11,827	Large WC gain (≥90th percentile: ≥2.42 cm/year)	RR 1st vs. 4th quartile	1.33 (1.12, 1.58) ^11^
				4527	Incident overweight/obesity	RR 1st vs. 4th quartile	1.20 (1.03, 1.40) ^11^
				4771	Incident obesity	RR 1st vs. 4th quartile	1.02 (0.85, 1.21) ^11^
Rohatgi 2017 [113]	Women’s Health Center and Obstetrics & Gynecology Clinic		MO, US	45	Gestational weight gain (kg)	Beta per 1% increase in UPF intake	1.3 (0.3, 2.4) ^12^
				45	Neonate thigh skinfold thickness (mm)	Beta per 1% increase in UPF intake	0.20 (0.005, 0.40) ^12^
				45	Neonate subscapular skinfold thickness (mm)	Beta per 1% increase in UPF intake	0.10 (0.02, 0.30) ^12^
				45	Neonate body fat percentage (%)	Beta per 1% increase in UPF intake	0.60 (0.04, 1.20) ^12^
Leone 2021 [114]	SUN	Females	Spain	3730	Gestational diabetes	OR 1st vs. 3rd tertile	1.10 (0.74, 1.64) ^13^
		Females < 30		2538	Gestational diabetes	OR 1st vs. 3rd tertile	0.89 (0.54, 1.46) ^13^
		Females ≥ 30		1192	Gestational diabetes	OR 1st vs. 3rd tertile	2.05 (1.03, 4.07) ^13^
Chang 2021 [115]	ALSPAC	Children	Britain	9020	BMI (kg/m^2^)/year	Beta 1st vs. 5th quintile	0.06 (0.04, 0.08) ^14^
				8078	Fat mass index (kg/m^2^)/year	Beta 1st vs. 5th quintile	0.03 (0.01, 0.05) ^14^
				8078	Lean mass index (kg/m^2^)/year	Beta 1st vs. 5th quintile	0.004 (−0.007, 0.01) ^14^
				8078	Body fat percentage (%)/year	Beta 1st vs. 5th quintile	0.004 (−0.05, 0.06) ^14^
Costa 2021 [116]	Pelotas-Brazil 2004 Birth Cohort	6–11-year-olds	Brazil	4231	Fat mass index (kg/m^2^)	Beta/100 g increase in UPF intake	0.09 (0.07, 0.10) ^15^
Srour 2019 [117]	Nutri-Net Santé	Adults ≥ 18	France	105,159	All CVD	HR per 10% increase in UPF	1.12 (1.05, 1.20) ^16^
				105,159	Coronary heart disease	HR per 10% increase in UPF	1.13 (1.02, 1.24) ^16^
				105,159	Cerebrovascular disease	HR per 10% increase in UPF	1.11 (1.01, 1.21) ^16^
Juul 2021 [118]	Framingham Offspring Cohort	Adults	US	3003	Overall CVD	HR per serving UPF/day	1.05 (1.02, 1.08) ^17^
				3003	CVD mortality	HR per serving UPF/day	1.09 (1.02, 1.16) ^17^
				3003	Incident hard CVD	HR per serving UPF/day	1.07 (1.03, 1.12) ^17^
				3003	Hard coronary heart disease	HR per serving UPF/day	1.09 (1.04, 1.15) ^17^
Zhong 2021 [119]	Prostate, Lung, Colorectal, and Ovarian Cancer Screening Trial	Adults aged 55–74 at baseline	US	91,891	CVD mortality	HR 1st vs. 5th quintile	1.50 (1.36, 1.64) ^18^
				91,891	Heart disease mortality	HR 1st vs. 5th quintile	1.68 (1.50, 1.87) ^18^
				91,891	Cerebrovascular disease mortality	HR 1st vs. 5th quintile	0.94 (0.76, 1.17) ^18^
Scaranni 2021 [120]	ELSA-Brasil	Civil servants aged 35–74 at baseline	Brazil	8754	Incident hypertension	OR 1st vs. 3rd tertile	1.20 (1.04, 1.40) ^19^
				8754	Change in SBP	Beta 1st vs. 3rd tertile	−0.37 (−1.05, 0.30) ^19^
				8754	Change in DBP	Beta 1st vs. 3rd tertile	0.19 (−0.28, 0.66) ^19^
Monge 2021 [121]	Mexican Teachers’ Cohort	Females aged ≥ 25 at baseline	Mexico	64,934	Incident hypertension	≤20% vs. >45% of energy from any UPF	0.99 (0.85, 1.15) ^20^
				64,934	Incident hypertension	≤20% vs. >45% of energy from liquid UPF	1.33 (1.09, 1.63) ^20^
				64,934	Incident hypertension	≤20% vs. >45% of energy from solid UPF	0.91 (0.82, 1.01) ^20^
Mendonca 2017 [122]	SUN	Middle-aged University graduates	Spain	14,790	Hypertension	HR 1st vs. 3rd tertile	1.23 (1.09, 1.38) ^21^
Llavero-Valero 2021 [123]	SUN	Middle-aged University graduates	Spain	20,060	T2DM	HR 1st vs. 3rd tertile	1.53 (1.06, 2.22) ^22^
Srour 2020 [124]	Nutri-Net Santé	Adults ≥ 18	France	104,707	T2DM	HR per 10% increase in UPF	1.15 (1.06, 1.25) ^23^
Zhang 2021 [125]	TCLSIH	Adults aged 18–90	China	16,168	NAFLD	HR 1st vs. 4th quartile	1.17 (1.07, 1.29) ^24^
Fiolet 2018 [126]	Nutri-Net Santé	Adults ≥ 18	France	104,980	All cancers	HR per 10% increase in UPF	1.12 (1.06, 1.18) ^25^
				104,980	Breast cancer	HR per 10% increase in UPF	1.11 (1.02, 1.22) ^25^
				104,980	Prostate cancer	HR per 10% increase in UPF	0.98 (0.83, 1.16) ^25^
				104,980	Colorectal cancer	HR per 10% increase in UPF	1.13 (0.92, 1.38) ^25^
Vasseur 2021 [127]	Nutri-Net Santé	Adults ≥ 18	France	105,832	IBD	RR 1st vs. 3rd tertile	1.32 (0.75, 2.34) ^26^
Narula 2021 [128]	PURE	Adults aged 35–70	21 low, middle, and high income countries	116,087	IBD	HR <1 vs. ≥5 servings UPF/day	1.82 (1.22, 2.72) ^27^
				116,087	Crohn’s disease	HR <1 vs. ≥5 servings UPF/day	4.50 (1.67, 12.13) ^27^
				116,087	Ulcerative Colitis	HR <1 vs. ≥5 servings UPF/day	1.46 (0.93, 2.28) ^27^
Schnabel 2018 [129]	Nutri-Net Santé	Adults	France	33,343	Irritable bowel syndrome	OR 1st vs. 4th quartile	1.24 (1.12, 1.38) ^28^
				33,343	Functional Constipation	OR 1st vs. 4th quartile	1.00 (0.87, 1.15) ^28^
				33,343	Functional diarrhoea	OR 1st vs. 4th quartile	0.94 (0.71, 1.26) ^28^
				33,343	Functional dyspepsia	OR 1st vs. 4th quartile	1.26 (1.07, 1.48) ^28^
Lo 2021 [130]	NHS, NHS II, HPFS	Adult health professionals	US	245,112	Crohn’s disease	HR 1st vs. 4th quartile	1.75 (1.29, 2.35) ^29^
				245,112	Ulcerative Colitis	HR 1st vs. 4th quartile	1.25 (0.97, 1.62) ^29^
Adjibade 2019 [131]	Nutri-Net Santé	Adults aged 18–86	France	26,730	Depressive symptoms	HR per 10% increase in UPF	1.21 (1.15, 1.27) ^30^
Gómez-Donoso 2020 [132]	SUN	Middle-aged University graduates	Spain	14,907	Incident depression	HR 1st vs. 4th quartile	1.41 (1.15, 1.73) ^31^
Rey-Garcia 2021 [133]	Seniors-ENRICA-1	Adult ≥ 60	Spain	1312	Renal function	OR 1st vs. 3rd tertile	1.75 (1.16, 2.64) ^32^
Zhang 2021 [134]	TCLSIH	Adults ≥ 18	China	18,444	Hyperuricemia	HR 1st vs. 4th quartile	1.21 (1.10, 1.33) ^33^
Leffa 2020 [135]	Impact of the “Ten Steps for Healthy Feeding of Children Younger Than Two Years” in Health Centers	Children	Porto Alegre, Brazil	308	Total cholesterol at age 6	Beta per 10% increase in UPF intake at age 3	0.07 (0.00, 0.15) *p* = 0.044 ^34^
				308	LDL-cholesterol at age 6	Beta per 10% increase in UPF intake at age 3	0.03 (−0.02, 0.09) ^34^
				308	HDL-cholesterol at age 6	Beta per 10% increase in UPF intake at age 3	0.01 (−0.01, 0.06) ^34^
				308	TAG at age 6	Beta per 10% increase in UPF intake at age 3	0.03 (0.00, 0.07) *p* = 0.034 ^34^
Donat-Vargas 2021 [136]	ENRICA	Adults > 60	Spain	895	Incident hypertriglyceridemia (≥150 mg/dL)	OR 1st vs. 3rd tertile	2.00 (1.04, 3.85) ^35^
				878	Low HDL-cholesterol (<40 in men or <50 mg/dL in women)	OR 1st vs. 3rd tertile	2.04 (1.22, 3.41) ^35^
				472	High LDL-cholesterol (>129 mg/dL)	OR 1st vs. 3rd tertile	0.95 (0.46, 1.97) ^35^
				895	Δtriglycerides (mg/dL)	Beta 1st vs. 3rd tertile	6.11 (1.30, 10.91) ^35^
				878	ΔHDL cholesterol (mg/dL)	Beta 1st vs. 3rd tertile	0.03 (−1.38, 1.44) ^35^
				472	ΔLDL cholesterol (mg/dL)	Beta 1st vs. 3rd tertile	−4.52 (−9.40, 0.36) ^35^
Borge 2021 [137]	Norwegian Mother, Father and Child Cohort Study	Mother and child	Norway	46,976	ADHD diagnosis at 8 years	RR per 1 SD increase in UPF	1.00 (0.93, 1.08) ^36^
				31,152	ADHD symptoms (absolute) at 8 years	Beta per 1 SD increase in UPF	0.38 (0.27, 0.49) ^36^ *
				31,152	ADHD symptoms (relative %) at 8 years	Beta per 1 SD increase in UPF	4.5 (3.3, 4.9) ^36^ *
Zhang 2021 [138]	TCLSIH	Adults ≥ 40	China	5409	Annual change in grip strength (kg per year)	Beta per 10% increase in UPF	−0.2955 (−0.4992, −0.0919) ^37^
				5409	Annual change in weight-adjusted grip strength (kg/kg per year)	Beta per 10% increase in UPF	−0.0043 (−0.0073, −0.0014) ^37^

Effect estimates are presented from adjusted models preceding the dietary adjustment, or where not performed, the adjusted model including dietary adjustment. OR, odds ratio; HR, hazard ratio; RR, relative risk; SD, standard deviation; ADHD, attention deficit hyperactivity disorder; CVD, cardiovascular disease; IHD, ischaemic heart disease; BMI, body mass index; T2DM, type 2 diabetes mellitus; UPF, ultra-processed food; SBP, systolic blood pressure; DBP, diastolic blood pressure; LDL, low-density lipoprotein; HDL, high-density lipoprotein; TAG, triacylglycerol; SUN, Seguimiento University of Navarra; NHANES III, Third National Health and Nutrition Examination Survey; CHNS, China Health and Nutrition Survey; Seniors-ENRICA 1, Seniors Study on Nutrition and Cardiovascular Risk in Spain; EPIC, European Prospective Investigation into Cancer and Nutrition; ELSA-Brazil, Brazilian Longitudinal Study of Adult Health; ALSPAC, Avon Longitudinal Study of Parents and Children; TCLSIH, Tianjin Chronic Low-grade Systemic Inflammation and Health; PURE, Prospective Urban Rural Epidemiology; NHS, Nurses’ Health Study 1986–2014); NHS II, Nurses’ Health Study II (1991–2017); HPFS, Health Professionals Follow-up Study; ^1^ Adjusted for sex, age, income level, education level, marital status, residence, BMI, physical activity level, smoking status, energy intake, alcohol intake, season of food records, first-degree family history of cancer or cardiovascular diseases and number of food records. ^2^ Adjusted for age, sex, marital status, physical activity, smoking status, snacking, special diet at baseline, BMI, total energy intake, alcohol consumption, family history of CVD, diabetes or hypertension at baseline, self-reported hypercholesterolaemia, baseline CVD, cancer or depression, education level and lifelong smoking stratified by recruitment period, deciles of age, sedentary index (sum of hours each day spent watching television, using a computer and driving) and television viewing. ^3^ Adjusted for age, sex, race/ethnicity, total energy intake, poverty level, education level, smoking status, physical activity and alcohol intake. ^4^ Adjusted for sex, age, energy intake, educational level, housing tenure, smoking, BMI, leisure-time physical activity, history of cancer, CVD, diabetes, hypertension and hyperlipidaemia and residence. ^5^ Adjusted for age, sex, marital status, educational level, physical activity, smoking status, alcohol consumption, energy intake and number of dietary records (overweight and obesity outcomes further adjusted for baseline BMI). ^6^ Adjusted for age, sex, marital status, educational status, physical activity, television watching, siesta sleep, smoking status, snacking between meals, following a special diet at baseline and baseline BMI. ^7^ Adjusted for age, sex and energy intake. ^8^ Adjusted for age, sex, study arm, follow-up time, educational level, marital status, smoking habits, type 2 diabetes prevalence, height and repeated measures of physical activity and sedentary behaviour. ^9^ Adjusted for age, sex, educational level, marital status, smoking, ex-drinker status, physical activity in the household and at leisure time, number of medications consumed per day and number of chronic diseases diagnosed by a doctor (chronic obstructive pulmonary disease/asthma, coronary heart disease, stroke, heart failure, osteoarthritis or depression). ^10^ Adjusted for age, sex, baseline BMI, educational level, physical activity, baseline alcohol intake, baseline smoking status and plausibility of dietary energy reporting. Overweight and obesity outcomes further adjusted for country/centre, follow-up time in years, smoking status at follow-up (instead of baseline smoking status) and for the modified relative Mediterranean diet score. ^11^ Adjusted for age, sex, colour/race, centre, income, school achievement, smoking and physical activity. Additionally for incident overweight/obesity and weight gain: baseline BMI. Additionally for waist gain: waist circumference at baseline. ^12^ Adjusted for maternal age, race, socioeconomic status, weight status, average daily energy intake, time spent in moderate physical activity and fat intake. ^13^ Adjusted for age, BMI, education, smoking status, physical activity, family history of diabetes, recruitment year, time between recruitment and the first pregnancy or gestational diabetes, number of pregnancies during follow-up, parity, multiple pregnancies, time spent watching TV, hypertension, following a nutritional therapy and energy intake. ^14^ Adjusted for age, baseline UPF, age*baseline UPF interaction term, child sex, race, birth weight, physical activity, quintiles of Index of Multiple Deprivation, the mother’s prepregnancy BMI, marital status, highest educational attainment, socioeconomic status and the child’s baseline total energy intake. ^15^ Adjusted for skin colour, maternal age and schooling, birthweight and sex, screen time and energy intake/expenditure ratio. ^16^ Adjusted for age, sex, energy intake, number of 24-h dietary records, smoking status, educational level, physical activity, BMI, alcohol intake and family history of CVD. ^17^ Adjusted for age, sex, education, smoking status, alcohol intake and physical activity. ^18^ Adjusted for age, sex, race, educational level, marital status, study centre, aspirin use, history of hypertension or diabetes, smoking status, alcohol consumption, BMI, physical activity and energy intake. ^19^ Adjusted for age, sex, colour or race, education, time since baseline and SBP/DBP/hypertension. ^20^ Adjusted for age, indigenous, internet access, insurance, family history of hypertension, menopausal status, smoking status and physical activity. ^21^ Adjusted for age, sex, physical activity, hours of TV watching, baseline BMI, smoking status, use of analgesics, following a special diet at baseline, family history of hypertension, hypercholesterolaemia and alcohol consumption. ^22^ Adjusted for age, sex, BMI, educational status, family history of diabetes, smoking status, snacking between meals, active and sedentary lifestyle score and following a special diet at baseline. ^23^ Adjusted for age, sex, educational level, baseline BMI, physical activity, smoking status, alcohol intake, number of 24-h dietary records, energy intake, Food Standards Agency nutrient profiling system dietary index score and family history of T2DM. ^24^ Adjusted for age, sex and BMI. ^25^ Adjusted for age, sex, energy intake without alcohol, number of 24-h dietary records, smoking status, educational level, physical activity, height, BMI, alcohol intake and family history of cancers (and for breast cancer outcome, additionally adjusted for menopausal status, hormonal treatment for menopause, oral contraception, and number of children). ^26^ Adjusted for age and sex. ^27^ Adjusted for age, sex, geographical region, education, alcohol intake, smoking status, BMI, total energy intake and location. ^28^ Adjusted for age, sex, income level, education level, marital status, residence, BMI, physical activity, smoking status, energy intake, season of food records and time between food record and functional gastrointestinal disorders questionnaire. ^29^ Adjusted for age, cohort and calendar year. ^30^ Adjusted for age, sex, BMI, marital status, educational level, occupational categories, household income per consumption unit, residential area, number of 24-h dietary records, inclusion month, energy intake without alcohol, alcohol intake, smoking status and physical activity. ^31^ Adjusted for sex, age and year of entrance to the cohort. ^32^ Adjusted for sex, age and total energy intake. ^33^ Adjusted for age, sex, BMI, smoking status, alcohol consumption status, education levels, employment status, household income, physical activity, depression symptoms, family history of disease (including cardiovascular disease, hypertension, hyperlipidaemia and diabetes), hypertension, hyperlipidaemia, diabetes and metabolic syndrome. ^34^ Adjusted for sex, group status in the early phase (intervention and control), family income, pre-pregnancy BMI, child birth weight and BMI z-scores at 3 years. ^35^ Adjusted for age and sex. ^36^ Crude model. ^37^ Adjusted for baseline age, sex and BMI. * 95% credible intervals.

**Table 3 nutrients-14-00023-t003:** Prospective studies adjusting for fat, added sugar/carbohydrate and sodium content.

Author, Year	Outcome	Method of Analysis	Diet Adjustment	Effect Estimate (95%CI)
Rico-Campa 2019 [103]	All-cause mortality	HR 1st vs. 4th quartile	SFA, sodium, added sugar and TFA	1.69 (1.12, 2.56)
Bonaccio 2021 [105]	All-cause mortality	HR 1st vs. 4th quartile	SFA, sodium, sugar, cholesterol and energy intake	1.28 (1.09, 1.49)
	CVD mortality	HR 1st vs. 4th quartile	SFA, sodium, sugar, cholesterol and energy intake	1.56 (1.19, 2.03)
	IHD/cerebrovascular mortality	HR 1st vs. 4th quartile	SFA, sodium, sugar, cholesterol and energy intake	1.33 (0.94, 1.90)
Beslay 2020 [106]	BMI change (kg/m^2^)	Beta per 10% increase in UPF	SFA, sodium, sugar and fibre	0.02 (0.01, 0.02)
	Overweight	HR per 10% increase in UPF	SFA, sodium, sugar and fibre	1.10 (1.08, 1.13)
	Obesity	HR per 10% increase in UPF	SFA, sodium, sugar and fibre	1.10 (1.06, 1.14)
Koniecnzna 2021 [109]	Total fat mass (z-score)	Beta per 10% increase in UPF	SFA, sodium, glycaemic index, TFA, alcohol and fibre	0.06 (0.03, 0.09)
	Visceral fat mass (z-score)	Beta per 10% increase in UPF	SFA, sodium, glycaemic index, TFA, alcohol and fibre	0.06 (0.01, 0.10)
	Android:gynoid fat ratio (z-score)	Beta per 10% increase in UPF	SFA, sodium, glycaemic index, TFA, alcohol and fibre	0.02 (−0.02, 0.07)
Srour 2019 [117]	All CVD	HR per 10% increase in UPF	SFA, sodium and sugar	1.13 (1.05, 1.20)
	Coronary heart disease	HR per 10% increase in UPF	SFA, sodium and sugar	1.14 (1.03, 1.26)
	Cerebrovascular disease	HR per 10% increase in UPF	SFA, sodium and sugar	1.12 (1.02, 1.22)
Zhong 2021 [119]	CVD mortality	HR 1st vs. 5th quintile	SFA, sodium and added sugar	1.48 (1.34, 1.63)
	Heart disease mortality	HR 1st vs. 5th quintile	SFA, sodium and added sugar	1.65 (1.47, 1.85)
	Cerebrovascular disease mortality	HR 1st vs. 5th quintile	SFA, sodium and added sugar	0.93 (0.74, 1.17)
Srour 2020 [124]	T2DM	HR per 10% increase in UPF	SFA, sodium, sugar and fibre	1.19 (1.09, 1.30)
Fiolet 2018 [126]	All cancers	HR per 10% increase in UPF	Lipids, sodium and carbohydrates	1.12 (1.07, 1.18)
	Breast cancer	HR per 10% increase in UPF	Lipids, sodium and carbohydrates	1.11 (1.01, 1.21)
	Prostate cancer	HR per 10% increase in UPF	Lipids, sodium and carbohydrates	0.98 (0.83, 1.16)
	Colorectal cancer	HR per 10% increase in UPF	Lipids, sodium and carbohydrates	1.16 (0.95, 1.42)
Chang 2021 [115]	BMI (kg/m^2^)/year	Beta 1st vs. 5th quintile	SFA, sodium, sugar and fibre	0.07 (0.04, 0.08)
	Fat mass index (kg/m^2^)/year	Beta 1st vs. 5th quintile	SFA, sodium, sugar and fibre	0.03 (0.01, 0.05)
	Lean mass index (kg/m^2^)/year	Beta 1st vs. 5th quintile	SFA, sodium, sugar and fibre	0.005 (−0.007, 0.010)
	Body fat percentage (%)/year	Beta 1st vs. 5th quintile	SFA, sodium, sugar and fibre	0.002 (−0.05, 0.05)

SFA, saturated fatty acids; TFA, trans fatty acids; HR, hazard ratio; CVD, cardiovascular disease; IHD, ischemic heart disease; BMI, body mass index; T2DM, type 2 diabetes mellitus.

**Table 4 nutrients-14-00023-t004:** Prospective studies adjusting for dietary pattern.

Author, Year	Outcome	Method of Analysis	Diet Adjustment	Effect Estimate (95%CI)
Schnabel 2019 [102]	All-cause mortality	HR per 10% increase in UPF	French dietary guidelines	1.14 (1.04, 1.27)
	All-cause mortality	HR per 10% increase in UPF	French dietary guidelines and Western dietary pattern	1.19 (1.05, 1.35)
Rico-Campa 2019 [103]	All-cause mortality	HR 1st vs. 4th quartile	Mediterranean dietary pattern	1.58 (1.10, 2.28)
Kim 2019 [104]	All-cause mortality	P-trend	Dietary quality score	*p*-trend only 0.001 ^1^
	CVD mortality	P-trend	Dietary quality score	*p*-trend only 0.540 ^1^
Bonaccio 2021 [105]	All-cause mortality	HR 1st vs. 4th quartile	Mediterranean dietary pattern	1.26 (1.09, 1.46)
	Other cause mortality (exc. CVD and cancer)	HR 1st vs. 4th quartile	Mediterranean dietary pattern	1.26 (0.94, 1.69)
	CVD mortality	HR 1st vs. 4th quartile	Mediterranean dietary pattern	1.58 (1.23, 2.03)
	IHD/cerebrovascular mortality	HR 1st vs. 4th quartile	Mediterranean dietary pattern	1.52 (1.10, 2.09)
	Cancer mortality	HR 1st vs. 4th quartile	Mediterranean dietary pattern	0.97 (0.77, 1.22)
Beslay 2020 [106]	BMI change (kg/m^2^)	Beta per 10% increase in UPF	Healthy and Western dietary patterns	0.02 (0.01, 0.02)
	Overweight	HR per 10% increase in UPF	Healthy and Western dietary patterns	1.10 (1.07, 1.13)
	Obesity	HR per 10% increase in UPF	Healthy and Western dietary patterns	1.11 (1.07, 1.15)
Li 2021 [108]	Overweight/obesity	OR none vs. ≥50 g/day	Traditional and modern dietary patterns	1.45 (1.21, 1.74) ^2^
	Central obesity	OR none vs. ≥50 g/day	Traditional and modern dietary patterns	1.50 (1.29, 1.74) ^2^
Koniecnzna 2021 [109]	Total fat mass (z-score)	Beta per 10% increase in UPF	Mediterranean dietary pattern adherence	0.06 (0.02, 0.09)
	Visceral fat mass (z-score)	Beta per 10% increase in UPF	Mediterranean dietary pattern adherence	0.06 (0.01, 0.10)
	Android:gynoid fat ratio (z-score)	Beta per 10% increase in UPF	Mediterranean dietary pattern adherence	0.02 (−0.02, 0.06)
Sandoval-Insausti 2020 [110]	Abdominal obesity	OR 1st vs. 3rd tertile	Mediterranean dietary pattern, fibre and very long chain omega-3 fatty acid intake	1.61 (1.01, 2.56)
Cordova 2021 [111]	Weight gain (kg)	Beta per 1SD increase in UPF/day	Mediterranean dietary pattern	0.118 (0.085, 0.151)
	Overweight/obesity	RR per 1SD increase in UPF/day	Mediterranean dietary pattern	1.05 (1.04, 1.06)
	Obesity	RR per 1SD increase in UPF/day	Mediterranean dietary pattern	1.05 (1.03, 1.07)
Leone 2021 [114]	Gestational diabetes pooled	OR 1st vs. 3rd tertile	Mediterranean dietary pattern	1.10 [0.74, 1.65)
	Gestational diabetes < 30	OR 1st vs. 3rd tertile	Mediterranean dietary pattern	0.89 [0.53, 1.47)
	Gestational diabetes ≥ 30	OR 1st vs. 3rd tertile	Mediterranean dietary pattern	2.06 (1.05, 4.06)
Costa 2021 [116]	Fat mass index (kg/m^2^)	Beta/100g daily increase in UPF intake	Unprocessed or minimally processed foods, processed culinary ingredients and processed foods intake	0.14 (0.13, 0.15)
Srour 2019 [117]	All CVD	HR per 10% increase in UPF	Healthy dietary pattern	1.11 (1.03, 1.19)
	Coronary heart disease	HR per 10% increase in UPF	Healthy dietary pattern	1.11 (1.00, 1.23) *p* = 0.04
	Cerebrovascular disease	HR per 10% increase in UPF	Healthy dietary pattern	1.10 (1.00, 1.20) *p* = 0.04
Juul 2021 [118]	Overall CVD	HR per serving UPF/day	Dietary Guidelines Adherence Index (DGAI) 2010	1.04 (1.01, 1.07)
	CVD mortality	HR per serving UPF/day	Dietary Guidelines Adherence Index (DGAI) 2010	1.09 (1.02, 1.16)
	Incident hard CVD	HR per serving UPF/day	Dietary Guidelines Adherence Index (DGAI) 2010	1.06 (1.02, 1.11)
	Hard coronary heart disease	HR per serving UPF/day	Dietary Guidelines Adherence Index (DGAI) 2010	1.09 (1.03, 1.15)
Zhong 2021 [119]	CVD mortality	HR 1st vs. 5th quintile	Healthy Eating Index (HEI) 2005	1.48 (1.35, 1.63)
	Heart disease mortality	HR 1st vs. 5th quintile	Healthy Eating Index (HEI) 2005	1.67 (1.49, 1.86)
	Cerebrovascular disease mortality	HR 1st vs. 5th quintile	Healthy Eating Index (HEI) 2005	0.94 (0.75, 1.16)
Llavero-Valero 2021 [123]	T2DM	HR 1st vs. 3rd tertile	Mediterranean dietary pattern	1.50 (1.02, 2.21)
Srour 2020 [124]	T2DM	HR per 10% increase in UPF	Healthy and Western dietary patterns	1.13 (1.04, 1.24)
Zhang 2021 [125]	NAFLD	HR 1st vs. 4th quartile	Healthy diet score	1.19 (1.08, 1.31) ^3^
Fiolet 2018 [126]	All cancers	HR per 10% increase in UPF	Western dietary pattern	1.12 (1.06, 1.18)
	Breast cancer	HR per 10% increase in UPF	Western dietary pattern	1.11 (1.02, 1.22)
	Prostate cancer	HR per 10% increase in UPF	Western dietary pattern	0.98 (0.83, 1.15)
	Colorectal cancer	HR per 10% increase in UPF	Western dietary pattern	1.13 (0.92, 1.38)
Vasseur 2021 [127]	IBD	RR 1st vs. 3rd tertile	Healthy dietary pattern	1.44 (0.70, 2.94) ^4^
Narula 2021 [128]	IBD	HR <1 vs. ≥5 servings UPF/day	Alternate Healthy Eating Index (AHEI) 2010	1.92 (1.28, 2.90)
	Crohn’s disease	HR <1 vs. ≥5 servings UPF/day	Alternate Healthy Eating Index (AHEI) 2010	4.90 (1.78, 13.45)
	Ulcerative Colitis	HR <1 vs. ≥5 servings UPF/day	Alternate Healthy Eating Index (AHEI) 2010	1.52 (0.96, 2.41)
Lo 2021 [130]	Crohn’s disease	HR 1st vs. 4th quartile	Alternate Healthy Eating Index (AHEI) 2010	1.70 (1.23, 2.35) ^5^
	Ulcerative Colitis	HR 1st vs. 4th quartile	Alternate Healthy Eating Index (AHEI) 2010	1.20 (0.91, 1.58) ^5^
Schnabel 2018 [129]	Irritable bowel syndrome	OR 1st vs. 4th quartile	French dietary guidelines	1.25 (1.12, 1.39)
	Functional Constipation	OR 1st vs. 4th quartile	French dietary guidelines	0.98 (0.85, 1.12)
	Functional diarrhoea	OR 1st vs. 4th quartile	French dietary guidelines	0.92 (0.69, 1.24)
	Functional dyspepsia	OR 1st vs. 4th quartile	French dietary guidelines	1.25 (1.05, 1.47)
Gómez-Donoso 2020 [132]	Incident depression	HR 1st vs. 4th quartile	Mediterranean dietary pattern	1.33 (1.07, 1.64) ^6^
Zhang 2021 [134]	Hyperuricemia	HR 1st vs. 4th quartile	Sweet, animal and healthy dietary patterns	1.17 (1.06, 1.30) ^7^
Donat-Vargas 2021 [136]	Incident hypertriglyceridemia (≥150 mg/dL)	OR 1st vs. 3rd tertile	Unprocessed or minimally processed food intake	2.66 (1.20, 5.90) ^8^
	Low HDL-cholesterol (<40 in men or <50 mg/dL in women)	OR 1st vs. 3rd tertile	Unprocessed or minimally processed food intake	2.23 (1.22, 4.05) ^8^
	High LDL-cholesterol (>129 mg/dL)	OR 1st vs. 3rd tertile	Unprocessed or minimally processed food intake	1.03 (0.43, 2.47) ^8^
	Δtriglycerides (mg/dL)	Beta 1st vs. 3rd tertile	Unprocessed or minimally processed food intake	6.87 (1.48, 12.27) ^8^
	ΔHDL cholesterol (mg/dL)	Beta 1st vs. 3rd tertile	Unprocessed or minimally processed food intake	0.13 (−1.46, 1.71) ^8^
	ΔLDL cholesterol (mg/dL)	Beta 1st vs. 3rd tertile	Unprocessed or minimally processed food intake	−2.03 (−7.86, 3.80) ^8^
Borge 2021 [137]	ADHD diagnosis at 8 years	RR per 1 SD increase in UPF	Child diet quality score at 3 years	1.07 (0.99, 1.18) ^9^
	ADHD symptoms (absolute) at 8 years	Beta per 1 SD increase in UPF	Child diet quality score at 3 years	0.25 (0.13, 0.38) ^9,^*
	ADHD symptoms (relative %) at 8 years	Beta per 1 SD increase in UPF	Child diet quality score at 3 years	3.0 (1.5, 4.5) ^9,^*
Zhang 2021 [138]	Change in grip strength (kg/year)	Beta per 10% increase in UPF	Healthy diet score	−0.3207 (−0.5281, −0.1133) ^10^
	Change in weight-adjusted grip strength (kg/kg/year)	Beta per 10% increase in UPF	Healthy diet score	−0.0046 (−0.0076, −0.0016) ^10^

OR, odds ratio; HR, hazard ratio; RR, relative risk; ADHD, attention deficit hyperactivity disorder; CVD, cardiovascular disease; IHD, ischemic heart disease; BMI, body mass index; T2DM, type 2 diabetes mellitus; IBD, inflammatory bowel disease; NAFLD, non-alcoholic fatty liver disease. ^1^ Further adjusted for body mass index, hypertension status, total cholesterol, and estimated glomerular filtration rate. ^2^ Further adjusted for fat intake, income, education, urbanisation, alcohol, smoking and physical activity. ^3^ Further adjusted for total energy intake, smoking status, alcohol drinking status, educational level, occupation, monthly household income, physical activity, family history of disease (including cardiovascular disease, hypertension, hyperlipidaemia and diabetes) and depressive symptoms. ^4^ Further adjusted for income level, education level, marital status, residence, BMI, physical activity, smoking status, hormonal contraception, number of 24-h dietary records and energy intake. ^5^ Further adjusted for race, family history of IBD, smoking, BMI, physical activity, total energy intake, regular NSAIDs use, oral contraceptives use, and menopausal hormone therapy. ^6^ Further adjusted for baseline BMI, total energy intake, physical activity, smoking status, marital status, living alone, employment status, working hours per week, health-related career, years of education and baseline self-perception of competitiveness, anxiety and dependence levels. ^7^ Further adjusted for energy intake. ^8^ Further adjusted for fibre intake, total energy intake, educational level, marital status, smoking status, BMI, physical activity, alcohol consumption, number of medications and number of chronic conditions. ^9^ Further adjusted for maternal pre-pregnancy BMI, maternal education, smoking and alcohol intake during pregnancy, maternal symptoms of depression and ADHD, maternal age, parity, child sex and childbirth quarter. ^10^ Further adjusted for smoking status, alcohol drinking status, education level, employment, monthly household income, physical activity, family history of disease (including CVD, hypertension, hyperlipidaemia and diabetes), depressive symptoms, hypertension, hyperlipidaemia, diabetes, total energy intake, dietary supplement use, total protein intake and milk intake. * 95% credible intervals.

**Table 5 nutrients-14-00023-t005:** Prospective cohort studies adjusting for total energy intake.

Author, Year	Outcome	Method of Analysis	Energy Adjustment	Effect
Schnabel 2019 [102]	All-cause mortality	HR per 10% increase in UPF	UPF as % weight + adjusted for TEI	1.15 (1.04, 1.27) ^1^
Rico-Campa 2019 [103]	All-cause mortality	HR 1st vs. 4th quartile	Energy-adjusted UPF + adjusted for TEI	1.62 (1.13, 2.33) ^2^
	Cardiovascular deaths	HR 1st vs. 4th quartile	Energy-adjusted UPF + adjusted for TEI	2.16 (0.92, 5.06) ^2^
	Cancer deaths	HR 1st vs. 4th quartile	Energy-adjusted UPF + adjusted for TEI	1.22 (0.70, 2.12) ^2^
Blanco-Rojo 2019 [91]	All-cause mortality	HR 1st vs. 4th quartile	UPF as % TEI	1.44 (1.01, 2.07) ^3^
Kim 2019 [104]	All-cause mortality	HR 1st vs. 4th quartile	UPF servings/day + adjusted for TEI	1.31 (1.09, 1.58) ^4^
	CVD mortality	HR 1st vs. 4th quartile	UPF servings/day + adjusted for TEI	1.10 (0.74, 1.67) ^4^
Romero Ferreiro 2021 [89]	All-cause mortality	HR per 10% increase in UPF	UPF as % TEI + adjusted for TEI	1.16 (1.06, 1.26) ^5^
Bonaccio 2021 [105]	All-cause mortality	HR 1st vs. 4th quartile	UPF as % weight + adjusted for TEI and energy content of UPFs	1.35 (1.15, 1.58) ^6^
	CVD mortality	HR 1st vs. 4th quartile	UPF as % weight + adjusted for TEI and energy content of UPFs	1.66 (1.28, 2.16) ^6^
	IHD/cerebrovascular mortality	HR 1st vs. 4th quartile	UPF as % weight + adjusted for TEI and energy content of UPFs	1.48 (1.05, 2.09) ^6^
	Cancer mortality	HR 1st vs. 4th quartile	UPF as % weight + adjusted for TEI	1.00 (0.80, 1.26) ^6^
	Other cause mortality	HR 1st vs. 4th quartile	UPF as % weight + adjusted for TEI	1.36 (1.01, 1.83) ^6^
Beslay 2020 [106]	BMI change (kg/m^2^)	Beta per 10% increase in UPF	UPF as % weight + adjusted for TEI	0.02 (0.01, 0.02) ^7^
	Overweight	HR per 10% increase in UPF	UPF as % weight + adjusted for TEI	1.11 (1.08, 1.14) ^7^
	Obesity	HR per 10% increase in UPF	UPF as % weight + adjusted for TEI	1.09 (1.05, 1.13) ^7^
Mendonça 2016 [107]	Overweight/obesity	HR 1st vs. 4th quartile	UPF servings/day + adjusted for TEI	1.27 (1.09, 1.49) ^8^
Li 2021 [108]	Overweight/obesity	OR none vs. ≥50 g/day	Absolute UPF g/day + adjusted for TEI	1.85 (1.58, 2.17) ^9^
	Central obesity	OR none vs. ≥50 g/day	Absolute UPF g/day + adjusted for TEI	2.04 (1.79, 2.33) ^9^
Koniecnzna 2021 [109]	Total fat mass (z-score)	Beta per 10% increase in UPF	UPF as % weight + adjusted for TEI	0.09 (0.06, 0.12) ^10^
	Visceral fat mass (z-score)	Beta per 10% increase in UPF	UPF as % weight + adjusted for TEI	0.09 (0.04, 0.13) ^10^
	Android:Gynoid fat ratio (z-score)	Beta per 10% increase in UPF	UPF as % weight + adjusted for TEI	0.04 (0.00, 0.08) *p* = 0.055 ^10^
Sandoval-Insausti 2020 [110]	Abdominal obesity	OR 1st vs. 3rd tertile	UPF as % TEI + adjusted for TEI	2.55 (1.04, 6.27) ^11^
Cordova 2021 [111]	Weight gain (kg)	Beta per 1SD increase in UPF/day	Energy-adjusted UPF	0.118 (0.085, 0.151) ^12^
	Overweight/obesity	RR per 1SD increase in UPF/day	Energy-adjusted UPF	1·05 (1.04, 1.06) ^12^
	Obesity	RR per 1SD increase in UPF/day	Energy-adjusted UPF	1·05 (1.03, 1.07) ^12^
Canhada 2020 [112]	Large weight gain (≥90th percentile: ≥1.68 kg/year)	RR 1st vs. 4th quartile	UPF as % TEI + adjusted for TEI	1.27 (1.07, 1.51) ^13^
	Large WC gain (≥90th percentile: ≥2.42 cm/year)	RR 1st vs. 4th quartile	UPF as % TEI + adjusted for TEI	1.36 (1.14, 1.61) ^13^
	Incident overweight/obesity	RR 1st vs. 4th quartile	UPF as % TEI + adjusted for TEI	1.22 (1.04, 1.42) ^13^
	Incident obesity	RR 1st vs. 4th quartile	UPF as % TEI + adjusted for TEI	1.02 (0.85, 1.21) ^13^
Rohatgi 2017 [113]	Gestational weight gain (kg)	Beta per 1% increase in UPF intake	UPF as % TEI + adjusted for TEI	1.3 (0.3, 2.4) ^14^
	Neonate thigh skinfold thickness (mm)	Beta per 1% increase in UPF intake	UPF as % TEI + adjusted for TEI	0.20 (0.005, 0.40) ^14^
	Neonate subscapular skinfold thickness (mm)	Beta per 1% increase in UPF intake	UPF as % TEI + adjusted for TEI	0.10 (0.02, 0.30) ^14^
	Neonate body fat percentage (%)	Beta per 1% increase in UPF intake	UPF as % TEI + adjusted for TEI	0.60 (0.04, 1.20) ^14^
Gomes 2021 [93]	Gestational weight gain 3rd trimester (kg)	Beta per 1% increase in UPF intake during 3rd trimester	UPF as % TEI	4.17 (0.55, 7.79) ^15^
	Gestational weight gain 2nd trimester (kg)	Beta per 1% increase in UPF intake in 2nd trimester	UPF as % TEI	−1.50 (−5.08, 2.08) ^15^
Leone 2021 [114]	Gestational diabetes pooled	OR 1st vs. 3rd tertile	Energy-adjusted UPF + adjusted for TEI	1.10 (0.74, 1.64) ^16^
	Gestational diabetes <30	OR 1st vs. 3rd tertile	Energy-adjusted UPF + adjusted for TEI	0.89 (0.54, 1.46) ^16^
	Gestational diabetes ≥30	OR 1st vs. 3rd tertile	Energy-adjusted UPF + adjusted for TEI	2.05 (1.03, 4.07) ^16^
Chang 2021 [115]	BMI (kg/m^2^)/year	Beta 1st vs. 5th quintile	UPF as % weight + adjusted for child’s TEI	0.06 (0.04, 0.08) ^17^
	Fat mass index (kg/m^2^)/year	Beta 1st vs. 5th quintile	UPF as % weight + adjusted for child’s TEI	0.03 (0.01, 0.05) ^17^
	Lean mass index (kg/m^2^)/year	Beta 1st vs. 5th quintile	UPF as % weight + adjusted for child’s TEI	0.004 (−0.007, 0.01) ^17^
	Body fat percentage (%)/year	Beta 1st vs. 5th quintile	UPF as % weight + adjusted for child’s TEI	0.004 (−0.05, 0.06) ^17^
	Weight (kg/year)	Beta 1st vs. 5th quintile	UPF as % weight + adjusted for child’s TEI	0.20 (0.11, 0.28) ^17^
	Waist circumference (cm/year)	Beta 1st vs. 5th quintile	UPF as % weight + adjusted for child’s TEI	0.17 (0.11, 0.22) ^17^
	BMI z-score	Beta 1st vs. 5th quintile	UPF as % weight + adjusted for child’s TEI	0.01 (0.003, 0.01) ^17^
	Fat mass (kg/year)	Beta 1st vs. 5th quintile	UPF as % weight + adjusted for child’s TEI	0.15 (0.08, 0.21) ^17^
	Lean mass (kg/year)	Beta 1st vs. 5th quintile	UPF as % weight + adjusted for child’s TEI	-0.04 (-0.11, 0.02) ^17^
Costa 2021 [116]	Fat mass index (kg/m^2^)	Beta/100 g increase in UPF intake	Absolute UPF g/day + adjusted for energy intake/expenditure ratio + TEI	0.05 (0.04, 0.06) ^18^
Vedovato 2021 [99]	BMI z-score age 10	Beta per 1 kcal/100 kcal/d increase in energy from UPF at age 4	UPF as % TEI at age 4	0.028 (0.006, 0.051) ^19^
	BMI z-score age 10	Beta per 1 kcal/100 kcal/d increase in energy from UPF at age 7	UPF as % TEI at age 7	0.014 (–0.007, 0.036) ^19^
	Enjoyment of food at age 7	Beta per 1 kcal/100 kcal/d increase in energy from UPF at age 4	UPF as % TEI at age 4	–0.002 (–0.021, 0.016) ^19^
	Food responsiveness at age 7	Beta per 1 kcal/100 kcal/d increase in energy from UPF at age 4	UPF as % TEI at age 4	0.017 (–0.001, 0.035) ^19^
	Emotional overeating at age 7	Beta per 1 kcal/100 kcal/d increase in energy from UPF at age 4	UPF as % TEI at age 4	0.010 (–0.006, 0.026) ^19^
	Emotional undereating at age 7	Beta per 1 kcal/100 kcal/d increase in energy from UPF at age 4	UPF as % TEI at age 4	0.007 (–0.012, 0.027) ^19^
	Satiety Responsiveness at age 7	Beta per 1 kcal/100 kcal/d increase in energy from UPF at age 4	UPF as % TEI at age 4	0.013 (–0.004, 0.029) ^19^
	Slowness in eating at age 7	Beta per 1 kcal/100 kcal/d increase in energy from UPF at age 4	UPF as % TEI at age 4	–0.015 (–0.035, 0.006) ^19^
	Food Fussiness at age 7	Beta per 1 kcal/100 kcal/d increase in energy from UPF at age 4	UPF as % TEI at age 4	0.026 (0.007, 0.045) ^19^
	Desire to Drink at age 7	Beta per 1 kcal/100 kcal/d increase in energy from UPF at age 4	UPF as % TEI at age 4	0.018 (–0.003, 0.039) ^19^
Costa 2019 [98]	△BMI age 4 to 8	Beta per 10% increase in UPF intake	UPF as % TEI at age 4	0.00 (–0.02, 0.01) ^20^
	△WC age 4 to 8	Beta per 10% increase in UPF intake	UPF as % TEI at age 4	0.07 (0.01, 0.13) ^20^
	△WHR age 4 to 8	Beta per 10% increase in UPF intake	UPF as % TEI at age 4	0.00 (0.00, 0.00) ^20^
	△Sum skinfolds age 4 to 8	Beta per 10% increase in UPF intake	UPF as % TEI at age 4	0.05 (−0.04, 0.15) ^20^
	Glucose (mmol/L)	Beta per 10% increase in UPF intake	UPF as % TEI at age 4	0.00 (−0.01, 0.00) ^20^
	Insulin (uU/mL)	Beta per 10% increase in UPF intake	UPF as % TEI at age 4	0.00 (−0.00, 0.01) ^20^
	HOMA-IR	Beta per 10% increase in UPF intake	UPF as % TEI at age 4	0.00 (−0.01, 0.01) ^20^
Srour 2019 [117]	All CVD	HR per 10% increase in UPF	UPF as % weight + adjusted for TEI	1.12 (1.05, 1.20) ^21^
	Coronary heart disease	HR per 10% increase in UPF	UPF as % weight + adjusted for TEI	1.13 (1.02, 1.24) ^21^
	Cerebrovascular disease	HR per 10% increase in UPF	UPF as % weight + adjusted for TEI	1.11 (1.01, 1.21) ^21^
Du 2021 [86]	Incident CAD	HR 1st vs. 4th quartile	Energy-adjusted UPF + adjusted for TEI	1.21 (1.06, 1.37) ^22^
Juul 2021 [118]	Overall CVD	HR per serving UPF/day	Energy-adjusted UPF + adjusted for TEI	1.05 (1.02, 1.08) ^23^
	CVD mortality	HR per serving UPF/day	Energy-adjusted UPF + adjusted for TEI	1.09 (1.02, 1.16) ^23^
	Incident hard CVD	HR per serving UPF/day	Energy-adjusted UPF + adjusted for TEI	1.07 (1.03, 1.12) ^23^
	Hard coronary heart disease	HR per serving UPF/day	Energy-adjusted UPF + adjusted for TEI	1.10 (1.04, 1.15) ^23^
Zhong 2021 [119]	CVD mortality	HR 1st vs. 5th quartile	Energy-adjusted UPF + adjusted for TEI	1.50 (1.36, 1.64) ^24^
	Heart disease mortality	HR 1st vs. 5th quartile	Energy-adjusted UPF + adjusted for TEI	1.68 (1.50, 1.87) ^24^
	Cerebrovascular disease mortality	HR 1st vs. 5th quartile	Energy-adjusted UPF + adjusted for TEI	0.94 (0.76, 1.17) ^24^
Scaranni 2021 [120]	Incident hypertension	OR 1st vs. 3rd tertile	UPF as % TEI + adjusted for TEI	1.23 (1.06, 1.44) ^25^
	Change in SBP	Beta 1st vs. 3rd tertile	UPF as % TEI + adjusted for TEI	–0.54 (–1.23, 0.15) ^25^
	Change in DBP	Beta 1st vs. 3rd tertile	UPF as % TEI + adjusted for TEI	0.08 (−0.39, 0.56) ^25^
Mendonça 2017 [122]	Hypertension	HR 1st vs. 3rd tertile	Energy-adjusted UPF + adjusted for TEI	1.21 (1.06, 1.37) ^26^
Rezende-Alves 2021 [148]	Hypertension	RR 1st vs. 5th quartile	UPF as % TEI	1.35 (1.01, 1.82) ^27^
Monge 2021 [121]	Incident hypertension	≤20% vs. >45% of energy from any UPF	UPF as % TEI + adjusted for TEI	0.98 (0.84, 1.14) ^28^
	Incident hypertension	≤20% vs. >45% of energy from liquid UPF	UPF as % TEI + adjusted for TEI	1.34 (1.10, 1.65) ^28^
	Incident hypertension	≤20% vs. >45% of energy from solid UPF	UPF as % TEI + adjusted for TEI	0.91 (0.82, 1.01) ^28^
Llavero-Valero 2021 [123]	T2DM	HR 1st vs. 3rd tertile	Energy-adjusted UPF + adjusted for TEI	1.52 (1.05, 2.22) ^29^
Srour 2020 [124]	T2DM	HR per 10% increase in UPF	UPF as % weight + adjusted for TEI	1.15 (1.06, 1.25) ^30^
Levy 2021 [88]	T2DM	HR per 10% increase in UPF	UPF as % weight + adjusted for TEI	1.20 (1.12, 1.29) ^31^
Zhang 2021 [125]	NAFLD	HR 1st vs. 4th quartile	UPF g/1000kcal + adjusted for TEI	1.19 (1.08, 1.31) ^32^
Fiolet 2018 [126]	All cancers	HR per 10% increase in UPF	UPF as % TEI + adjusted for TEI (exc. Alcohol)	1.12 (1.06, 1.18) ^33^
	Breast cancer	HR per 10% increase in UPF	UPF as % TEI + adjusted for TEI (exc. Alcohol)	1.11 (1.02, 1.22) ^33^
	Prostate cancer	HR per 10% increase in UPF	UPF as % TEI + adjusted for TEI (exc. Alcohol)	0.98 (0.83, 1.16) ^33^
	Colorectal cancer	HR per 10% increase in UPF	UPF as % TEI + adjusted for TEI (exc. Alcohol)	1.13 (0.92, 1.38) ^33^
Vasseur 2021 [127]	IBD	RR 1st vs. 3rd tertile	UPF as % weight + adjusted for TEI	1.44 (0.70, 2.94) ^34^
Narula 2021 [128]	IBD	HR <1 vs. ≥5 servings UPF/day	UPF servings/day + adjusted for TEI	1.82 (1.22, 2.72) ^35^
	Crohn’s disease	HR <1 vs. ≥5 servings UPF/day	UPF servings/day + adjusted for TEI	4.50 (1.67, 12.13) ^35^
	Ulcerative Colitis	HR <1 vs. ≥5 servings UPF/day	UPF servings/day + adjusted for TEI	1.46 (0.93, 2.28) ^35^
Schnabel 2018 [129]	Irritable bowel syndrome	OR 1st vs. 4th quartile	UPF as % weight + adjusted for TEI	1.24 (1.12, 1.38) ^36^
	Functional Constipation	OR 1st vs. 4th quartile	UPF as % weight + adjusted for TEI	1.00 (0.87, 1.15) ^36^
	Functional diarrhoea	OR 1st vs. 4th quartile	UPF as % weight + adjusted for TEI	0.94 (0.71, 1.26) ^36^
	Functional dyspepsia	OR 1st vs. 4th quartile	UPF as % weight + adjusted for TEI	1.26 (1.07, 1.48) ^36^
Lo 2021 [130]	Crohn’s disease	HR 1st vs. 4th quartile	UPF as % TEI + adjusted for TEI	1.70 (1.23, 2.35) ^37^
	Ulcerative Colitis	HR 1st vs. 4th quartile	UPF as % TEI + adjusted for TEI	1.20 (0.91, 1.58) ^37^
Adjibade 2019 [131]	Depressive symptoms	per 10% increase in UPF	UPF as % weight + adjusted for TEI	1.21 (1.15, 1.27) ^38^
Gómez-Donoso 2020 [132]	Incident depression	HR 1st vs. 4th quartile	Energy-adjusted UPF + adjusted for TEI	1.33 (1.07, 1.64) ^39^
Rey-Garcia 2021 [133]	Renal function	OR 1st vs. 3rd tertile	UPF as % TEI + adjusted for TEI	1.75 (1.16, 2.64) ^40^
Zhang 2021 [134]	Hyperuricemia	HR 1st vs. 4th quartile	UPF servings/day + adjusted for TEI	1.17 (1.06, 1.30) ^41^
Leffa 2020 [135]	Total cholesterol at age 6	Beta per 10% increase in UPF intake at age 3	UPF as % TEI + adjusted for TEI	0.07 (0.00, 0.14) *p* = 0.046 ^42^
	LDL-cholesterol at age 6	Beta per 10% increase in UPF intake at age 3	UPF as % TEI + adjusted for TEI	0.03 (–0.03, 0.09) ^42^
	HDL-cholesterol at age 6	Beta per 10% increase in UPF intake at age 3	UPF as % TEI + adjusted for TEI	0.01 (–0.02, 0.05) ^42^
	TAG at age 6	Beta per 10% increase in UPF intake at age 3	UPF as % TEI + adjusted for TEI	0.04 (0.01, 0.07) *p* = 0.024 ^42^
Rauber 2015 [95]	ΔTotal cholesterol from 3–4 to 7–8	Beta per 1% increase in energy from UPF	UPF as % TEI + adjusted for TEI at age 7–8	0.430 (0.008, 0.853) ^43^
	ΔLDL cholesterol from 3–4 to 7–8	Beta per 1% increase in energy from UPF	UPF as % TEI + adjusted for TEI at age 7–8	0.369 (0.005, 0.733) ^43^
	ΔnHDL cholesterol from 3–4 to 7–8	Beta per 1% increase in energy from UPF	UPF as % TEI + adjusted for TEI at age 7–8	0.319 (−0.059, 0.697) ^43^
	ΔTriglycerides from 3–4 to 7–8	Beta per 1% increase in energy from UPF	UPF as % TEI + adjusted for TEI at age 7–8	−0.465 (−0.955, 0.025) ^43^
	ΔHDL cholesterol from 3–4 to 7–8	Beta per 1% increase in energy from UPF	UPF as % TEI + adjusted for TEI at age 7–8	0.125 (−0.026, 0.277) ^43^
Donat-Vargas 2021 [136]	Incident hypertriglyceridemia (≥150 mg/dL)	OR 1st vs. 3rd tertile	UPF as % TEI + adjusted for TEI	2.21 (1.09, 4.49) ^44^
	Low HDL-cholesterol (<40 in men or <50 mg/dL in women)	OR 1st vs. 3rd tertile	UPF as % TEI + adjusted for TEI	2.04 (1.18, 3.53) ^44^
	High LDL-cholesterol (>129 mg/dL)	OR 1st vs. 3rd tertile	UPF as % TEI + adjusted for TEI	1.13 (0.52, 2.46) ^44^
	Δtriglycerides (mg/dL)	Beta 1st vs. 3rd tertile	UPF as % TEI + adjusted for TEI	6.23 (1.26, 11.21) ^44^
	ΔHDL cholesterol (mg/dL)	Beta 1st vs. 3rd tertile	UPF as % TEI + adjusted for TEI	0.02 (−1.45, 1.49) ^44^
	ΔLDL cholesterol (mg/dL)	Beta 1st vs. 3rd tertile	UPF as % TEI + adjusted for TEI	−3.43 (−8.60, 1.74) ^44^
Machado Azeredo 2020 [101]	Wheeze at age 11	OR 1st vs. 5th quintile of UPF at age 6	UPF as % TEI + adjusted for TEI and TEI:EEI	0.78 (0.51, 1.19) ^45^
	Asthma at age 11	OR 1st vs. 5th quintile of UPF at age 6	UPF as % TEI + adjusted for TEI and TEI:EEI	0.83 (0.59, 1.17) ^45^
	Mild/moderate asthma at age 11	OR 1st vs. 5th quintile of UPF at age 6	UPF as % TEI + adjusted for TEI and TEI:EEI	0.63 (0.34, 1.17) ^45^
	Severe Asthma at age 11	OR 1st vs. 5th quintile of UPF at age 6	UPF as % TEI + adjusted for TEI and TEI:EEI	0.94 (0.54, 1.65) ^45^
Borge 2021 [137]	ADHD diagnosis at 8 years	RR per 1 SD increase in UPF	UPF as % TEI	1.07 (0.99, 1.18) ^46^
	ADHD symptoms (absolute) at 8 years	Beta per 1 SD increase in UPF	UPF as % TEI	0.25 (0.13, 0.38) ^46,^*
	ADHD symptoms (relative) at 8 years	Beta per 1 SD increase in UPF	UPF as % TEI	3.0 (1.5, 4.5) ^46,^*
Zhang 2021 [138]	Change in grip strength (kg/year)	Beta per 10% increase in UPF	UPF as % weight + adjusted for TEI	−0.3207 (−0.5281, −0.1133) ^47^
	Change in weight-adjusted grip strength (kg/kg/year)	Beta per 10% increase in UPF	UPF as % weight + adjusted for TEI	−0.0046 (−0.0076, −0.0016) ^47^

Energy-adjusted UPF via the residual method. TEI, total energy intake; TEI:EEI, energy intake:expenditure ratio; OR, odds ratio, HR, hazard ratio; RR, relative risk; ADHD, attention deficit hyperactivity disorder; CVD, cardiovascular disease; IHD, ischemic heart disease; HOMA-IR, Homeostatic Model Assessment for Insulin Resistance; BMI, body mass index; T2DM, type 2 diabetes mellitus; IBD, inflammatory bowel disease; NAFLD, non-alcoholic fatty liver disease; UPF, ultra-processed food; SBP, systolic blood pressure; DBP, diastolic blood pressure; LDL, low-density lipoprotein; HDL, high-density lipoprotein; nHDL, non-high-density lipoprotein; TAG, triacylglycerol; WC, waist circumference; WHR, waist-to-hip ratio. ^1^ Adjusted for sex, age, income level, education level, marital status, residence, BMI, physical activity level, smoking status, energy intake, alcohol intake, season of food records, first-degree family history of cancer or cardiovascular diseases and number of food records. ^2^ Adjusted for age, sex, marital status, physical activity, smoking status, snacking, special diet at baseline, BMI, total energy intake, alcohol consumption, family history of CVD, diabetes or hypertension at baseline, self-reported hypercholesterolaemia, baseline CVD, cancer or depression, education level and lifelong smoking stratified by recruitment period, deciles of age, sedentary index (sum of hours each day spent watching television, using a computer and driving) and television viewing. ^3^ Adjusted for age, sex, education level, living alone, smoking status, former drinker, physical activity, time watching television, time devoted to other sedentary activities, number of medications per day and specific chronic conditions diagnosed by a physician (chronic respiratory disease, coronary heart disease, stroke, heart failure, osteoarthritis, cancer and depression). ^4^ Adjusted for age, sex, race/ethnicity, total energy intake, poverty level, education level, smoking status, physical activity and alcohol intake. ^5^ Adjusted for age, sex, BMI, physical activity, alcohol intake, smoking status and total energy intake. ^6^ Adjusted for sex, age, energy intake, educational level, housing tenure, smoking, BMI, leisure-time physical activity, history of cancer, CVD, diabetes, hypertension and hyperlipidaemia and residence. ^7^ Adjusted for age, sex, marital status, educational level, physical activity, smoking status, alcohol consumption, energy intake and number of dietary records (overweight and obesity outcomes further adjusted for baseline BMI). ^8^ Adjusted for age, sex, marital status, educational status, physical activity, energy intake, television watching, siesta sleep, smoking status, snacking between meals, following a special diet at baseline, baseline BMI and consumption of fruit and vegetables. ^9^ Adjusted for age, sex and energy intake. ^10^ Adjusted for age, sex, study arm, follow-up time, educational level, marital status, smoking habits, type 2 diabetes prevalence, height and repeated measures of total energy intake, physical activity and sedentary behaviour. ^11^ Adjusted for age, sex, educational level, marital status, smoking, ex-drinker status, physical activity in the household and at leisure time, number of medications consumed per day, number of chronic diseases diagnosed by a doctor (chronic obstructive pulmonary disease/asthma, coronary heart disease, stroke, heart failure, osteoarthritis or depression) and total energy intake, Mediterranean dietary pattern, fibre and very long chain omega-3 fatty acid intake. ^12^ Adjusted for age, sex, baseline BMI, educational level, physical activity, baseline alcohol intake, baseline smoking status and plausibility of dietary energy reporting and for the modified relative Mediterranean diet score. Overweight and obesity outcomes further adjusted for country/centre, follow-up time in years and smoking status at follow-up (instead of baseline smoking status). ^13^ Adjusted for age, sex, colour/race, centre, income, school achievement, smoking, physical activity and energy intake. Additionally for incident overweight/obesity and weight gain: baseline BMI. Additionally for waist gain: waist circumference at baseline. ^14^ Adjusted for maternal age, race, socioeconomic status, weight status, average daily energy intake, time spent in moderate physical activity and fat intake. ^15^ Adjusted for cohort and potential confounding variables, with *p* < 0·25. ^16^ Adjusted for age, BMI, education, smoking status, physical activity, family history of diabetes, recruitment year, time between recruitment and the first pregnancy or gestational diabetes, number of pregnancies during follow-up, parity, multiple pregnancies, time spent watching TV, hypertension, following a nutritional therapy, and energy intake. ^17^ Adjusted for age, baseline UPF, age*baseline UPF interaction term, child sex, race, birth weight, physical activity, quintiles of Index of Multiple Deprivation, the mother’s prepregnancy BMI, marital status, highest educational attainment, socioeconomic status and the child’s baseline total energy intake. ^18^ Adjusted for skin colour, maternal age and schooling, birthweight and sex, screen time, total energy intake and energy intake/expenditure ratio. ^19^ Adjusted for maternal age, maternal education and BMI before pregnancy, exclusive breast-feeding for the first 6 months, child physical exercise and daily screen time. For appetitive behaviours, additionally parental concerns, and for BMI, additionally child BMI z-score at 4 years old. ^20^ Adjusted for sex, group status in the early phase (intervention and control), pre-pregnancy BMI, birth weight, breastfeeding, family income, maternal schooling and total screen duration. ^21^ Adjusted for age, sex, energy intake, number of 24-h dietary records, smoking status, educational level, physical activity, BMI, alcohol intake, and family history of CVD. ^22^ Adjusted for age, sex, race, centre and total energy intake. ^23^ Adjusted for age, sex, education, smoking status, alcohol intake, total energy intake and physical activity. ^24^ Adjusted for age, sex, race, educational level, marital status, study centre, aspirin use, history of hypertension or diabetes, smoking status, alcohol consumption, BMI, physical activity and energy intake. ^25^ Adjusted for age, sex, colour or race, education, time since baseline, SBP/DBP/hypertension, physical activity, smoking, alcohol consumption, sodium intake and total energy intake. ^26^ Adjusted for age, sex, physical activity, hours of TV watching, baseline BMI, smoking status, use of analgesics, following a special diet at baseline, family history of hypertension, hypercholesterolemia, alcohol consumption, total energy intake, olive oil intake and consumption of fruits and vegetables. ^27^ Adjusted for age, marital status, skin colour, per capita income, physical activity, smoking, obesity, family history of hypertension and previous diagnosis of T2DM, hypercholesterolaemia and hypertriglyceridaemia. ^28^ Adjusted for age, indigenous, internet access, insurance, family history of hypertension, menopausal status, smoking status, physical activity, total energy intake and multivitamin intake. ^29^ Adjusted for age, sex, BMI, educational status, family history of diabetes, smoking status, snacking between meals, active and sedentary lifestyle score, following a special diet at baseline and total energy intake. ^30^ Adjusted for age, sex, educational level, baseline BMI, physical activity, smoking status, alcohol intake, number of 24-h dietary records, total energy intake, Food Standards Agency nutrient profiling system dietary index score, and family history of T2DM. ^31^ Adjusted for age, sex, ethnicity, family history of T2DM, Index of Multiple Deprivation, physical activity level, current smoking status and total energy intake. ^32^ Adjusted for age, sex, BMI, healthy diet score, total energy intake, smoking status, alcohol drinking status, educational level, occupation, monthly household income, physical activity, family history of disease (including cardiovascular disease, hypertension, hyperlipidaemia and diabetes) and depressive symptoms. ^33^ Adjusted for age, sex, total energy intake without alcohol, number of 24-h dietary records, smoking status, educational level, physical activity, height, BMI, alcohol intake, family history of cancers (and for breast cancer outcome, additionally adjusted for menopausal status, hormonal treatment for menopause, oral contraception and number of children). ^34^ Adjusted for age, sex, income level, education level, marital status, residence, BMI, physical activity, smoking status, hormonal contraception, number of 24-h dietary records, healthy dietary pattern and total energy intake. ^35^ Adjusted for age, sex, geographical region, education, alcohol intake, smoking status, BMI, total energy intake, and location. ^36^ Adjusted for age, sex, income level, education level, marital status, residence, BMI, physical activity, smoking status, total energy intake, season of food records and time between food record and functional gastrointestinal disorders questionnaire. ^37^ Adjusted for age, cohort, calendar year, AHEI-2010, race, family history of IBD, smoking, BMI, physical activity, total energy intake, regular NSAIDs use, oral contraceptives use and menopausal hormone therapy. ^38^ Adjusted for age, sex, BMI, marital status, educational level, occupational categories, household income per consumption unit, residential area, number of 24-h dietary records, inclusion month, total energy intake without alcohol, alcohol intake, smoking status and physical activity. ^39^ Adjusted for age, sex, year of entrance to the cohort, Mediterranean diet, baseline BMI, total energy intake, physical activity, smoking status, marital status, living alone, employment status, working hours per week, health-related career, years of education and baseline self-perception of competitiveness, anxiety and dependence levels. ^40^ Adjusted for age, sex and total energy intake. ^41^ Adjusted for age, sex, BMI, smoking status, alcohol consumption status, education levels, employment status, household income, physical activity, depression symptoms, family history of disease (including cardiovascular disease, hypertension, hyperlipidaemia and diabetes), hypertension, hyperlipidaemia, diabetes, metabolic syndrome, total energy intake and dietary patterns (sweet, animal and healthy patterns). ^42^ Adjusted for sex, group status in the early phase (intervention and control), family income, pre-pregnancy BMI, childbirth weight, BMI z-scores at 3 years, total energy and total fat intake at 3 years. ^43^ Adjusted for sex, group, birth weight, family income, maternal schooling, BMI-for-age z-scores and total energy intake at 7–8 years. ^44^ Adjusted for age, sex, fibre intake, total energy intake, educational level, marital status, smoking status, BMI, physical activity, alcohol consumption, number of medications and number of chronic conditions. ^45^ Adjusted for TEI and TEI:EEI. ^46^ Adjusted for child diet quality score using Diet Quality Index at 3 years, maternal pre-pregnancy BMI, maternal education, smoking and alcohol intake during pregnancy, maternal symptoms of depression and ADHD, maternal age, parity, child sex and child birth quarter. ^47^ Adjusted for baseline age, sex, BMI, smoking status, alcohol drinking status, education level, employment, monthly household income, physical activity, family history of disease (including CVD, hypertension, hyperlipidaemia, and diabetes), depressive symptoms, hypertension, hyperlipidaemia, diabetes, total energy intake, healthy diet score, dietary supplement use, total protein intake and milk intake. * 95% credible intervals.

## Data Availability

Not applicable.

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
