# Peer review of "The Role of Diet Quality in Mediating the Association between Ultra-Processed Food Intake, Obesity and Health-Related Outcomes: A Review of Prospective Cohort Studies"

_nutrients, 2021, doi:10.3390/nu14010023_

Round 1

Reviewer 1 Report

This is an important, well-written and thorough paper. The review raises an important issue: The UPF impact on health risk is not entirely mediated by dietary nutrients or patterns. FI only have a few minor commnts and queries.

  1. Theoretically, UPF intake is perceived as part of diet quality. For that reason, inserting 'components' after diet quality in the title could be considered.
  2. Certain foods are considered UPF and others not, based on what may be regarded as sociological or environmental considerations, not on nutritional or technological criteria. For instance, pre-packaged bread is UPF, but not unpackaged bread; the same for cheese, as well as for ready-to-eat vs home-prepared foods. This could be touched upon. Furthermore, the items that are included in the UPF group may slightly vary from one study to the other, no? Another point for discussion.
  3. The search terms and process would have to be described for the review of prospective studies included in the review.
  4. The title of section 6.4 is the same as for 6.5; and section 6.6 is repeated.
  5. Keywords: Ultra-processed FOOD; NOVA CLASSIFICATION

Author Response

We thank this reviewer for their helpful comments. We have responded to the points raised within the reviewer’s text for clarity.

This is an important, well-written and thorough paper. The review raises an important issue: The UPF impact on health risk is not entirely mediated by dietary nutrients or patterns. FI only have a few minor commnts and queries. 

Response: Thank you for the positive comments.

1. Theoretically, UPF intake is perceived as part of diet quality. For that reason, inserting 'components' after diet quality in the title could be considered.

Response: The authors agree that UPF could be considered as being part of diet quality. However, the authors feel the current title is most informative, as diet quality is typically considered in terms of nutrients and diet patterns, as discussed in the review.

2. Certain foods are considered UPF and others not, based on what may be regarded as sociological or environmental considerations, not on nutritional or technological criteria. For instance, pre-packaged bread is UPF, but not unpackaged bread; the same for cheese, as well as for ready-to-eat vs home-prepared foods. This could be touched upon. Furthermore, the items that are included in the UPF group may slightly vary from one study to the other, no? Another point for discussion.

Response: Thank you for raising this point. Indeed, pre-packaged vs. artisanal bread have different classifications in NOVA. However, these are still considered to be a reflection of the industrial processes used to mass produce UPFs (ready meals) which are not possible at home, vs. homemade processes (home-prepared meals).

We have added to line 811 in the limitations to reflect your point: “Similar foods can be classed as ultra-processed (e.g. pre-packaged bread) or processed (e.g. artisanal bread), which may result in the misclassification of foods with the dietary assessment methods used.”

3. The search terms and process would have to be described for the review of prospective studies included in the review.

Response: To maintain the flow of the manuscript, the following has been added to the supplementary materials:

Review process

Papers were included in the review if: they were prospective studies examining the impact of UPF intake (defined by the NOVA classification) on any health-related outcome, and also performed any form of diet or energy adjustment in their modelling analyses. Papers were excluded if they were cross-sectional in nature, retrospective, or did not include any form of diet or energy adjustment in their modelling to determine the association between UPF and the health-related outcome.

Prospective studies were obtained for the review via searches on PubMed for “ultraprocessed”, and “(ultraprocessed) AND ((Prospective) OR (Longitudinal))”. All item titles were reviewed and relevant abstracts were screened. Full papers were then accessed and included or excluded from the review based on the above criteria. Searches through references of systematic and narrative reviews of UPF and health outcomes were also performed, as well as searches through references of prospective studies.

All prospective studies examining the impact of UPF intake (defined by the NOVA classification) on any health-related outcome, whether or not they performed any form of diet or energy adjustment in their modelling analyses from the search process, were cited in the manuscript.”

and a sentence has been added on line 209 in the manuscript: “(the search process and criteria for the review is detailed in the Supplementary Materials).”

4. The title of section 6.4 is the same as for 6.5; and section 6.6 is repeated.

Response:

Thank you for pointing this out. It appears to have been an error when formatting the manuscript into the Nutrients layout for submission after the authors proofed the draft manuscript. These have now been updated to:

line 557:

“6.5. Dietary adjustments that explain the association between UPF intake and health-related outcomes”

line 690:

“6.7. Prospective studies reporting mediation analyses”

5. Keywords: Ultra-processed FOOD; NOVA CLASSIFICATION

Response: The keywords have been updated to include these changes.

Reviewer 2 Report

Comments to the Authors of manuscript number: nutrients-1511804 entitled “The role of diet quality in mediating the association between ultra-processed food intake, obesity and health-related outcomes: a review of prospective cohort studies”.

The authors have presented a review about ultra-processed foods and its link with and poor health. Many people consume food with fat (typically saturated fat), carbohydrate (typically sugar) and sodium leading to serious health problem. 

  1. The manuscript appears well written and well structured.
  2. The abstract concisely provides the overview.
  3. The authors make big attention in details.
  4. The information provided is easily accessible by the reader.
  5. Overall, this review provides a valuable synthesis of the current state of our knowledge regarding ultra-processed different type of food and obesity and obesity-related death, cancer, cardiovascular disease or type 2 diabetes.
  6. It is worth to add the figure presenting the way by which different criteria were given. This which allow to exclude and include the paper during searching.
  7. It is worth to present keyword used to search. It should be presented as a part of methods.
  8. Authors presented very recently published literature. Predominant number of papers are published within 5 years.

Author Response

We thank this reviewer for their helpful comments. We have responded to the points raised within the reviewer’s text for clarity.

Comments to the Authors of manuscript number: nutrients-1511804 entitled “The role of diet quality in mediating the association between ultra-processed food intake, obesity and health-related outcomes: a review of prospective cohort studies”.

The authors have presented a review about ultra-processed foods and its link with and poor health. Many people consume food with fat (typically saturated fat), carbohydrate (typically sugar) and sodium leading to serious health problem.  

  1. The manuscript appears well written and well structured.
  2. The abstract concisely provides the overview.
  3. The authors make big attention in details.
  4. The information provided is easily accessible by the reader.
  5. Overall, this review provides a valuable synthesis of the current state of our knowledge regarding ultra-processed different type of food and obesity and obesity-related death, cancer, cardiovascular disease or type 2 diabetes.

Response. We thank the reviewer for their positive comments.

6. It is worth to add the figure presenting the way by which different criteria were given. This which allow to exclude and include the paper during searching.

Response: A PRISMA flow diagram is typically reserved for a systematic review, which this manuscript is not. We have outlined the search methods used and criteria for the review: to maintain the flow of the manuscript, the following has been added to the supplementary materials:

Review process

Papers were included in the review if: they were prospective studies examining the impact of UPF intake (defined by the NOVA classification) on any health-related outcome, and also performed any form of diet or energy adjustment in their modelling analyses. Papers were excluded if they were cross-sectional in nature, retrospective, or did not include any form of diet or energy adjustment in their modelling to determine the association between UPF and the health-related outcome.

Prospective studies were obtained for the review via searches on PubMed for “ultraprocessed”, and “(ultraprocessed) AND ((Prospective) OR (Longitudinal))”. All item titles were reviewed and relevant abstracts were screened. Full papers were then accessed and included or excluded from the review based on the above criteria. Searches through references of systematic and narrative reviews of UPF and health outcomes were also performed, as well as searches through references of prospective studies.

All prospective studies examining the impact of UPF intake on any health-related outcome, whether or not they performed any form of diet or energy adjustment in their modelling analyses from the search process, were cited in the manuscript.”

and a sentence has been added on line 209 in the manuscript: “(the search process and criteria for the review is detailed in the Supplementary Materials).”

7. It is worth to present keyword used to search. It should be presented as a part of methods.

Response: Keywords used to perform the review search have been provided in the supplementary materials, as detailed above in response to point 6.

8. Authors presented very recently published literature. Predominant number of papers are published within 5 years.

Response: Thank you for acknowledging this strength of the manuscript